# AGENTREWARDBENCH: Evaluating Automatic Evaluations of Web Agent Trajectories

**Xing Han Lù**[1][2]  **Amirhossein Kazemnejad**[*][2]
**Nicholas Meade**[1][2]  **Arkil Patel**[1][2]  **Dongchan Shin**[2]  **Alejandra Zambrano**[2]
**Karolina Stańczak**[1][2]  **Peter Shaw**[4]  **Christopher J. Pal**[2][5][6][7]  **Siva Reddy**[1][2][5][7]
[*]Core contributor  [1]McGill University  [2]Mila Quebec AI Institute  [4]Google DeepMind
[5]Canada CIFAR AI Chair  [6]Polytechnique Montréal  [7]ServiceNow Research
xing.han.lu@mail.mcgill.ca; siva.reddy@mila.quebec

## Abstract

Web agents enable users to perform tasks on web browsers through natural language interaction. Evaluating web agents trajectories is an important problem, since it helps us determine whether the agent successfully completed the tasks. Rule-based methods are widely used for this purpose, but they are challenging to extend to new tasks and may not always recognize successful trajectories. We may achieve higher accuracy through human evaluation, but the process would be substantially slower and more expensive. Automatic evaluations with LLMs may avoid the challenges of designing new rules and manually annotating trajectories, enabling faster and cost-effective evaluation. However, it is unclear how effective they are at evaluating web agents. To this end, we propose AGENTREWARD-BENCH, the first benchmark to assess the effectiveness of LLM judges for evaluating web agents. AGENTREWARDBENCH contains 1302 trajectories across 5 benchmarks and 4 LLMs. Each trajectory in AGENTREWARD-BENCH is reviewed by an expert, who answers questions pertaining to the success, side effects, and repetitiveness of the agent. Using our benchmark, we evaluate 12 LLM judges and find that no single LLM excels across all benchmarks. We also find that the rule-based evaluation used by common benchmarks tends to underreport the success rate of web agents, highlighting a key weakness of rule-based evaluation and the need to develop more flexible automatic evaluations. We release the benchmark at: https://agent-reward-bench.github.io

## 1 Introduction

Giving a Large Language Model (LLM) access to a web browser unlocks an entirely new capability paradigm: beyond interacting with a user through a chat interface, such models can interact with the online world to complete tasks similar to how a human would. The promise of a new paradigm has motivated the design of LLMs to control interfaces such as web browsers, starting from earlier foundation models such as *ACT-1* (Adept, 2022) to the more recent OpenAI *Operator* (OpenAI, 2025) and Claude *Computer use* (Anthropic, 2024a), showing promising results in real-world tasks (Zhou et al., 2024).

To measure the progress of web agents, a well-designed benchmark should compile a collection of realistic tasks across diverse websites. As illustrated in Figure 1, a user may ask the agent to locate a Classifieds listing for a Google Pixel phone and submit an offer via a comment. Inside a dedicated environment (e.g., a self-hosted Classifieds site), the web agent would complete the task by filling the search bar, identifying the correct listing, and writing a comment to show interest in purchasing the item. To determine if the agent successfully completed the request, we need to automatically evaluate the agent's chosen actions – known as *trajectories* – using a set of rules uniquely designed for the task of finding a Pixel phone on Classifieds. As expected, rule-based evaluation is time-consuming for

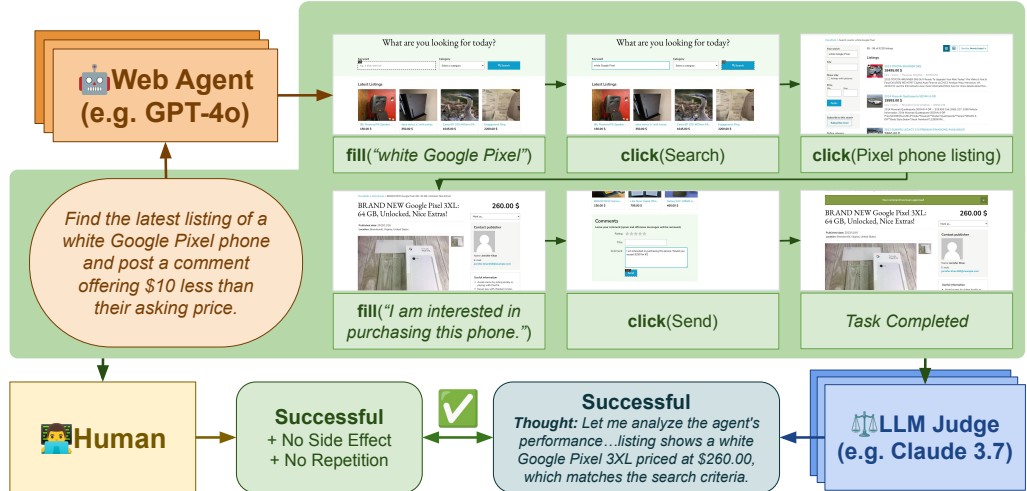

Figure 1: Example from AGENTREWARDBENCH, where an LLM judge evaluates a web agent's trajectory on VisualWebArena (Koh et al., 2024). The benchmark compares judgments against expert annotations to determine the effectiveness of the judge for evaluating web agents.

experts to design, and may not cover every successful scenario (e.g., what if the agent finds a different but valid listing?). It is also possible for an expert to annotate the trajectories, but it would be slow and expensive to scale across many web agents. This brings us to the following questions: **Given a web agent trajectory, can an LLM decide if it is successful? If so, how do we determine which LLM is the most capable at evaluating web agents?**

Past works have shown that LLMs can be used as judges to evaluate the output of LLM chatbots (Zheng et al., 2023). More recently, LLM judges have been used for automatically evaluating trajectories from web agents (Pan et al., 2024; Murty et al., 2025; Trabucco et al., 2025). With highly accurate automatic evaluation methods, we can measure the progress of web agents on new sets of tasks, use them to synthesize trajectories for finetuning smaller models, and design reward models that can be used in a reinforcement learning (RL) setting. However, it remains unclear whether current automatic evaluators, whether rule-based or LLM-based, can predict the success of a trajectory in a way that reflects expert judgment.

To address this problem, we introduce AGENTREWARDBENCH (§3), a benchmark for determining the capability of an LLM at evaluating web agents (see Figure 1). It consists of 1300 trajectories produced by 4 popular LLM agents on 5 diverse web environments, ranging from common tasks like online shopping and posting on a forum, to highly specialized requests in professional environments, such as updating task schedules on IT task management platforms. Each trajectory is labeled by expert annotators to determine whether the agent successfully completed the task, caused unintended side effects, or entered cycles of repetitive actions. Using this benchmark, we evaluate both existing and novel LLM judges (§4) alongside rule-based evaluation. We find that rule-based methods, which are used as the official automatic evaluation by environment-based benchmarks, severely underestimate the capabilities of agents and do not reflect how experts define success (§5). We further provide an in-depth analysis (§6) that highlights the weaknesses of existing LLMs when used as judges. Overall, we believe AGENTREWARDBENCH can be used to enable better automatic evaluation and reward modeling for web agents.

## 2   Related Works

**Web Agents and Environments**   Designing agents that can automatically navigate user interfaces has been a long standing problem; earlier approaches employed program-based heuristics (St. Amant & Zettlemoyer, 2000), whereas later works on web navigation focus on training reinforcement learning (RL) models (Gur et al., 2018; Humphreys et al., 2022),

language models (Nakano et al., 2021; Gur et al., 2023; Deng et al., 2023) and multimodal models (Shaw et al., 2023; Lù et al., 2024; Zheng et al., 2024). To measure the advancements in web agents, various benchmarks have been proposed, with initial works proposing simplified environments (Shi et al., 2017; Liu et al., 2018) and subsequent iterations focusing on specific tasks like web shopping (Yao et al., 2022). More recent benchmarks focus on designing realistic environments that cover commonly used websites (Zhou et al., 2024; Koh et al., 2024) as well as specialized environments (Drouin et al., 2024; Boisvert et al., 2025).

**LLM Judges** Zheng et al. (2023) proposed using LLMs to predict human preferences of dialogue completion for chat models. They show that a *GPT-4*-based judge achieves over 80% agreement with human votes on the task of selecting better completions between models pairs. Subsequently, Agent-as-a-Judge (AaJ) was proposed by Zhuge et al. (2024) to extend LLM judges to coding agents, leveraging intermediate feedback from the environment. He et al. (2024) extend the idea by using LLMs to judge trajectories from web agents, allowing them to determine task completion without human annotators, resulting in a high correlation with humans on a private subset of trajectories. To determine the quality of automatic judgments, Pan et al. (2024) evaluate four LLM judges using trajectories from a *GPT-4* agent on WebArena tasks, and find that the best judge achieves 80.6% accuracy against the rule-based evaluator from WebArena. Unlike prior works on LLM judges, we design AGENTREWARDBENCH with trajectories from several LLM agents on diverse web benchmarks, where each one is annotated by human experts on multiple dimensions. By following a human-focused approach similar to Lambert et al. (2024), we ensure that LLM judges are evaluated against expert preferences on a wide range of scenarios.

**Trajectory Synthesis** Leveraging web environments that can be created and reset without real-world impact, recent works started to explore generating trajectories without human supervision. Leveraging LLM judges and LLM-generated tasks, trajectory synthesis can be used to bootstrap agent-judge training loops (Murty et al., 2024; 2025), to create contrastive pairs (Putta et al., 2024) for direct preference optimization (Rafailov et al., 2023), or as training data to finetune a base model (Lai et al., 2024; Patel et al., 2024; Trabucco et al., 2025). Although all the methods leverage an LLM judge, they lack a clear way of directly determining the quality of judgments, instead relying on the downstream performance improvement to validate their approach. To this end, AGENTREWARDBENCH enables researchers to choose the most appropriate LLM judge for a category of web tasks based on their effectiveness at evaluating web agents.

## 3  AGENTREWARDBENCH

In this work, we introduce AGENTREWARDBENCH, a benchmark designed to assess the capabilities of LLM judges for evaluating web agents (§3.1). We curate 5 diverse web environments and tasks (§3.2) in order to collect trajectories from web agents based on 4 LLMs (§3.3). For each trajectory, a team of expert annotators carefully reviews the screenshots, actions, and the agent's reasoning chains before labeling them as either successful or unsuccessful, alongside other auxiliary labels (see Figure 2). Finally, we evaluate LLM judges (Table 1) by comparing their predictions with expert annotations to determine their effectiveness for automatic evaluation.

### 3.1  Assessment Framework

**Trajectory Definition** Let $o_i$ be an observation of a browser at time step $i$, $a_i$ be an action that can be executed on a webpage through a browser navigation engine $B$ such that $o_{i+1} = B(o_i, a_i)$, and $r_i$ be the reasoning for choosing the action. We define a web agent trajectory as the sequence $\mathcal{T} = \{o_1, (r_1, a_1), o_2, (r_2, a_2), \ldots, o_{n-1}, (r_{n-1}, a_{n-1}), o_n\}$ where $o_n$ is the final observation in the trajectory. Each observation contains a screenshot of the browser $s_i$, the Document Object Model (DOM) tree representation of the browser, and an accessibility (A11Y) tree rendered from the DOM tree. For the observation to be useful for an LLM agent, we need a representation function $R$ that produces $p_i = R(o_1, r_1, a_1, \ldots, o_i)$, which can be used as an input for an LLM. If the agent is multimodal, $o_i$ would include screenshots; otherwise, it would be a textual representation of the page (e.g., accessibility

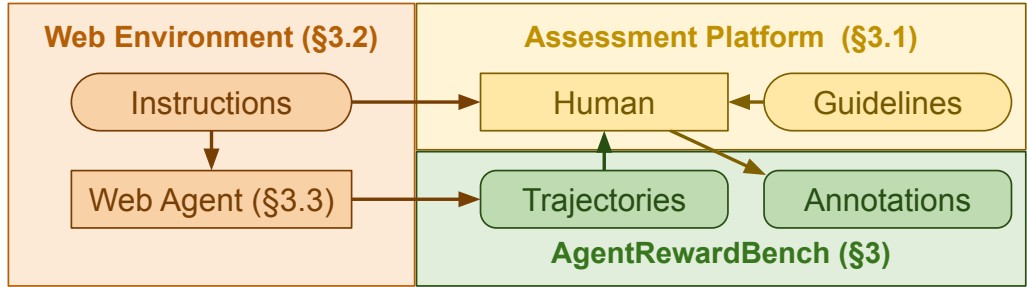

Figure 2: AGENTREWARDBENCH creation process. We first collect trajectories from LLM agents inside web environments using instructions from several benchmarks. Then, the trajectories are reviewed by expert annotators, who indicate if the trajectory is successful, led to side effects, and contains repetition cycles. Finally, we use the annotated trajectories to evaluate LLM judges.

tree). Then, $p_i$ is given to a language model to produce a completion $c_i = \text{LM}(p_i)$, or $c_i = \text{VLM}(p_i, s_i)$ in the case of a multimodal LLM. The completion is parsed by an execution function $E$ to produce $(a_i, r_i) = E(c_i)$.

**Annotation Design**   For each trajectory, an expert annotator reviews a goal $g$ and sequence $\{s_1, (r_1, a_1), \ldots, s_{n-1}, (r_{n-1}, a_{n-1}), s_n\}$ in order to answer questions $\mathcal{Q} = \{q_1, \ldots, q_m\}$. We consider the answers produced, $\mathcal{A}^* = \{a_1^*, \ldots, a_m^*\}$, as the ground truth annotations for the trajectory, which indicate whether the agent successfully completed $g$. To collect $\mathcal{A}^*$, we use the following $\mathcal{Q}$ in the annotation guidelines:

1. **Success**: Was the sequence of actions successful in achieving the goal?
2. *Side Effect*: Did the agent perform unnecessary actions that could lead to unintended side effects?
3. *Repetition Cycle*: Did the agent loop through a sequence of actions that did not make progress towards the goal?

Agreement with respect to **success** is the primary criterion with which we evaluate LLM judges. The remaining can be useful as auxiliary criteria for detecting issues ahead of time. For example, if an agent purchases several irrelevant products when the user only requested one, then the trajectory would be flagged for side effects, independent of task success. A judge can also indicate the presence of a cycle, for example, if the agent repeatedly clicks on a disabled button. Both signals can be used to penalize the agent during training or steer it to another action at inference.

**Annotation Setup**   The team of annotators consisted of 6 experts with a deep understanding of the tasks and environments through their research on web agents. They used a custom-built user interface that displays each trajectory with screenshots, actions, and reasoning. Rating disagreements were resolved by annotators discussing among themselves until clear annotations can be produced for ambiguous trajectories. Moreover, the annotators also have access to the environment and accessibility trees when screenshots are insufficient.

**Judge Model**   Given a goal $g$, trajectory $\mathcal{T}$ and questions $\mathcal{Q}$, a judge model returns a judgment $\hat{\mathcal{A}}$, which is an estimate of $\mathcal{A}^*$. We can use $\hat{\mathcal{A}}$ to derive a reward in RL or to automatically evaluate web agents when $\mathcal{A}^*$ is unavailable. To implement the judge, we need a judge-specific function $R_j$ that produces a representation of the trajectory, $p = R_j(o_1, r_1, a_1, \ldots, o_n)$. $R_j=$ can vary substantially, ranging from a simple list of actions $a_1, \ldots, a_{n-1}$, to using another LLM to process the observation history. We describe judges used in previous works and introduce a simplified judge in Section 4 and provide supplementary details in Appendix A.3.

### 3.2 Tasks and Environments

We select 5 benchmarks designed to evaluate web agents inside dedicated environments and real websites, including general-purpose (Zhou et al., 2024), vision-focused (Koh et al.,

2024), real-world (Yoran et al., 2024), and enterprise-oriented (Drouin et al., 2024; Boisvert et al., 2025) tasks. In total, we curate 351 unique tasks across 8 environments and 66 websites, which we separate into 51 development and 300 test tasks (details in Appendix A.1).

**WebArena (WA; Zhou et al. 2024)**  This benchmark comprises 6 self-hosted websites covering a wide range of domains: customer relationship management, map navigation, online encyclopedia, shopping site, social forum, and software development collaboration platform. Each environment is derived from real open-source projects that develop self-hosted environments for both commercial and personal usage. Each task consists of a textual goal that requires a good understanding of one or multiple environments to complete.

**VisualWebArena (VWA; Koh et al. 2024)**  To complement WebArena's text-based goals, we also include VisualWebArena (VWA), a benchmark focusing on tasks that require visual reasoning to complete. For instance, a user may include an image alongside the goal, or the task could be designed to only be solved if the agent selects an item with a unique visual characteristic (e.g., purchasing a TV with the widest bezel). VWA also introduces a new online marketplace environment (Classifieds).

**AssistantBench (AB; Yoran et al. 2024)**  In addition to the self-hosted environments, we consider trajectories resulting from agent execution on real-world websites. This benchmark defines tasks that require navigating the internet, starting from a search engine. Since the test set is private, we use the validation set, which consists of 33 unique tasks.

**WorkArena (Work; Drouin et al. 2024) and WorkArena++ (Wk++; Boisvert et al. 2025)** To increase the diversity of tasks relevant to professional environments, we incorporate WorkArena (Boisvert et al., 2025), a benchmark of 18 basic tasks on ServiceNow,[1] a software-as-a-service platform for professional workflows in the information technology (IT), human resources, and customer management domains. WorkArena++ introduces tasks with greater complexity, requiring planning and reasoning to correctly complete multiple sub-tasks. Including this alongside WorkArena allows us to evaluate judges on a wider range of task difficulty. We focus on the Level 2 tasks since Level 3 is too challenging for current agents.

### 3.3  Web Agents Design

To collect trajectories on the 5 benchmarks, we design web agents using two models from major commercial providers and two open-weight LLMs.

**LLM backbones**  On the commercial side, we use OpenAI's *GPT-4o*[2] (Hurst et al., 2024) and Anthropic's *Claude 3.7 Sonnet* (Anthropic, 2024b). They are the flagship models of their respective providers, both of which offer computer-use agents powered by their LLMs, namely OpenAI *Operator* (OpenAI, 2025) and Anthropic Claude's *Computer use* (Anthropic, 2024a). We select two leading open-weights LLMs to complement the commercial LLMs: *Llama-3.3-70B* (Grattafiori et al., 2024) and *Qwen2.5-VL* (Bai et al., 2025). In both cases, we choose the instruction-tuned variant, which have undergone post-training for tool-use or UI navigation. Moreover, since Llama-3.3 is a text-only model, it was excluded from VisualWebArena, which requires image-based reasoning.

**Agent Platform**  By default, each LLM backbone receives an input processed by a representation function $R$ and generates a completion $c_i$. Then, $c_i$ is interpreted as an action by an execution function $E$. To implement $E$, we use AgentLab and BrowserGym (Chezelles et al., 2025), an ecosystem for designing web agents using LLMs (details in Appendix A.1).

**Trajectory Annotations and Splits**  We collect a total of 1302 trajectories from our 4 LLM-based web agents across five benchmarks. Based on the task split (§3.2), 196 trajectories are in the development split and 1106 are in the test split (details in A.2). The annotators follow the process described in Section 3.1 to label all trajectories, producing a total of 3906 binary annotations. To assess agreement between annotators, we annotated the GPT-4o agent's

---

[1] https://developer.servicenow.com
[2] We use the version gpt-4o-2024-11-20

| Category | Judge | Overall | | | AB | VWA | WA | Work | Wk++ |
| | | Precision | Recall | F1 | | | Precision | | |
| Official | *Rule-based*[*] | 83.8 | 55.9 | 67.1 | 25.0 | 85.2 | 79.0 | 100.0 | 83.3 |
| Existing | AER-C | 67.7 | 71.9 | 69.7 | 83.3 | 56.0 | 68.8 | 100.0 | 66.7 |
| | AER-V | 67.6 | 71.5 | 69.5 | 83.3 | 61.2 | 67.6 | 96.4 | 59.3 |
| | NNetNav | 52.5 | 82.4 | 64.1 | 20.8 | 54.5 | 54.3 | 77.3 | 43.2 |
| Ours (A) | Claude 3.7 S. | 68.8 | 81.6 | 74.7 | 87.5 | 61.0 | 69.3 | 85.0 | 66.7 |
| | GPT-4o | **69.8** | 83.1 | 75.9 | 77.8 | 63.0 | 70.2 | 94.6 | 63.0 |
| | GPT-4o Mini | 61.5 | 86.1 | 71.7 | 80.0 | 57.9 | 63.5 | 84.2 | 49.4 |
| | Llama 3.3 | 67.7 | 79.0 | 72.9 | 75.0 | 59.6 | 68.2 | 94.3 | 62.7 |
| | Qwen2.5-VL | 64.3 | 89.8 | 75.0 | 72.7 | 59.3 | 63.6 | 87.2 | 60.3 |
| Ours (S) | Claude 3.7 S. | 69.4 | 76.3 | 72.7 | 71.4 | 64.8 | 69.3 | 85.3 | 66.7 |
| | GPT-4o | 68.1 | 80.3 | 73.7 | 77.8 | 60.7 | 69.9 | 93.8 | 59.6 |
| | GPT-4o Mini | 64.5 | 78.3 | 70.8 | 80.0 | 57.4 | 66.9 | 90.3 | 54.8 |
| | Qwen2.5-VL | 64.5 | 86.1 | 73.7 | 70.0 | 58.5 | 62.9 | 93.8 | 64.4 |

Table 1: Judge performance for predicting success, measured with *precision* (§4.2). We report *recall* and *F1* as auxiliary scores. We examine two variants of the simplified judge: one with the final accessibility tree (A), and the other with the final screenshot (S). *\*Rule-based evaluation are included for reference.*

trajectory on WebArena with a second annotator. We obtained an inter-annotator agreement of 89.3% on success, indicating a high level of consistency among annotators.

## 4 LLM judges for web tasks

### 4.1 Judge implementations

We consider two existing implementations of LLM judges for web agents, *Agent Eval Refine* (AER; Pan et al. 2024) and *NNetNav* (Murty et al., 2025), and introduce a **simplified judge** that simultaneously predicts success, side effects, and repetition. In *Agent-as-a-Judge* (Zhuge et al., 2024), the method assumes the judge can interact with the environment after the agent finishes executing actions, which isn't feasible when the environment state cannot be preserved or shareable across agents. Other LLM judge variants were proposed (He et al., 2024; Putta et al., 2024; Lai et al., 2024; Trabucco et al., 2025), but our three judge implementations cover major strategies for representing trajectories.

**AER (Pan et al., 2024)**   The judge in this framework takes as input the sequence of agent thoughts and actions alongside the final browser state, which is either passed to a vision-enabled model as a screenshot (AER-V) or as a caption generated by a captioner model (AER-C). Then, the judge outputs its reasoning before predicting success or failure. For both the judge and captioner, we implement this method using GPT-4o, which is an overall stronger model than the GPT-4 (Achiam et al., 2023) model originally used.

**NNetNav (Murty et al., 2025)**   In this work, a Llama 3.1 70B judge receives a summary of changes across all observations and has to give a rating between 1 (worst) and 5 (best) after providing the thought process; the rating is binarized by thresholding at 4, based on the original implementation. To generate summaries, an LLM is used to describe the change between two observations based on the accessibility trees instead of screenshots. We use Llama 3.3 70B (Al-Dahle, 2024), an improved version of the original backbone.

**Simplified judge (ours)**   We propose a simplified design for our judge. First, it directly answers the three questions asked to the annotators. This allows it to return multiple labels within a single completion. Then, we decouple the system prompt and reasoning chain from the final state representation, allowing the judge to receive either the accessibility tree or the screenshot. This differs from AER, which requires a vision-enabled model, and NNetNav, which requires a long-context model capable of receiving multiple accessibility trees. Our

| A11Y | Screen | Success | | | Side Effect | | | Repetition | | |
|---|---|---|---|---|---|---|---|---|---|---|
| | | P | R | F1 | P | R | F1 | P | R | F1 |
| ✓ | ✓ | 62.1 | 81.7 | 70.6 | 6.5 | 31.9 | 10.8 | 92.5 | 16.8 | 28.4 |
| ✓ | ✗ | 61.5 | 86.1 | 71.7 | 7.2 | 70.8 | 13.0 | 78.6 | 46.4 | 58.3 |
| ✗ | ✓ | 64.5 | 78.3 | 70.8 | 6.6 | 31.9 | 11.0 | 92.3 | 18.5 | 30.8 |
| ✗ | ✗ | 60.7 | 73.9 | 66.7 | 7.2 | 76.4 | 13.2 | 78.1 | 59.1 | 67.3 |

Table 2: Ablation study of our *GPT-4o mini* judge, measured in precision (P), recall (R), and F1. We consider how including accessibility trees and screenshots in the input affects the predictions.

method is compatible with both multimodal and text-only LLMs and does not require a separate LLM to caption the screenshot or summarize changes across observations.

## 4.2 Evaluation

To evaluate LLM judges, we use the *precision* score, which is the ratio of true positives over all predicted positives (true + false positives). The metric is a good fit for rejection finetuning (RFT), where we are interested in increasing the number of true positives (actual successful trajectories) while reducing the number of false positives (failed trajectories added to the dataset due to poor LLM judgments). For reward modeling, we also want to prioritize true positives since they are the primary signals for many RL algorithms, while false positives should be minimized to avoid introducing noise to the loss function. Moreover, *recall* and *F1* benefit from minimizing false negatives, which is useful for improving sample efficiency by reducing the number of valid trajectories removed; we report them as auxiliary metrics.

## 4.3 Judge Performance

In Table 1, we provide an overview of the performance of judges across benchmarks using the metrics defined in Section 4.2. We find that *GPT-4o* and *Claude 3.7 Sonnet*-based simplified judges achieve higher precision compared to prior approaches, indicating that removing the internal LLMs for captioning or summarizing changes does not hinder their capabilities. Notably, no judge consistently stands out across benchmarks, highlighting the importance of selecting an appropriate LLM backbone based on the nature of the task.

**Low precision limits existing judges** We notice that no judge achieves above 70% precision, which means that 30% of trajectories are erroneously marked as successful. This severely limits the usefulness of the judges for downstream applications, such as using the filtered trajectories for finetuning an agent, as the agent will learn to generate incorrect trajectories for a substantial portion of the tasks. This indicates LLM judges are currently not a reliable way of assessing the true capabilities of agents. Consequently, judges will need to achieve higher precision before they are useful for automatic evaluation, which also affects their downstream utility for methods like RFT and RL.

**Official rule-based evaluation underestimates success** Similar to LLM judges, the rule-based evaluation used by benchmarks can be compared with expert annotations. Since they use task-specific configurations to determine success, they may reject successful trajectories due to inconsequential differences. For instance, in WebArena, if a user asks "What's the closest national park to the largest city in Maine?", the agent may reply: "The closest national park to Portland [...] is Acadia National Park". Rule-based evaluation considers it unsuccessful since the configuration requires it to exactly match "Acadia National Park". As a result, the rule-based approach achieves a *recall* of 55.9%, indicating a higher rate of false negatives compared to LLM judges. Overall, a substantial precision gap exists between rule-based methods and LLM judges, but rule-based methods severely underestimate the true performance of web agents, highlighting the need for more flexible automatic evaluation.

**Impact of Input Representation** Browser screenshots represent an intuitive state for humans, but LLMs may need more than vision alone, as screenshots miss page structure and hidden attributes found in accessibility trees. To investigate the impact of different representations, we ablate our *GPT-4o-mini* simplified judge in Table 2. We observe that

| Agent | Human | | | GPT-4o Judge | | | Rule-based | | |
|---|---|---|---|---|---|---|---|---|---|
| | VWA | WA | Wk++ | VWA | WA | Wk++ | VWA | WA | Wk++ |
| Claude 3.7 S. | 28.3 | 55.1 | 18.4 | 34.8 | 64.1 | 20.7 | 23.9 | 30.8 | 8.1 |
| GPT-4o | 35.9 | 42.3 | 18.4 | 47.8 | 50.0 | 11.5 | 17.4 | 25.6 | 4.6 |
| Llama 3.3 | 0.0 | 22.4 | 9.2 | 0.0 | 27.6 | 5.8 | 0.0 | 18.4 | 3.5 |
| Qwen2.5-VL | 21.7 | 33.3 | 13.8 | 34.8 | 52.6 | 14.9 | 17.4 | 29.5 | 11.5 |

Table 3: Success Rate of web agents measured by expert annotators, GPT-4o Judge (with accessibility tree) and rule-based evaluation on various benchmarks (§3.2). Results by environment are in Table 6.

only including screenshots achieves a high precision for success and repetition, whereas only including accessibility trees allows higher recall. Surprisingly, including both accessibility trees and screenshots yields a lower performance than including only the screenshot, indicating that more information distracts rather than assists the judge.

## 5 Revisiting how we evaluate task success rate

One of the core applications of LLM judges is to estimate the success rate on a web navigation benchmark, which is useful in scenarios where there are no dedicated functions to calculate the rule-based success rate, which is the standard evaluation for many web agent benchmarks. However, rule-based approaches may not always agree with experts. In Table 3, we compare the success rate calculated from expert annotations, rule-based evaluation, and a GPT-4o judge with accessibility trees.

**Rule-based evaluation does not reflect expert-defined success rates** We notice a stark difference between the judge and rule-based approach: whereas the LLM judge tends to overestimate the success rate of every agent (with two exceptions in WorkArena++), rule-based methods consistently underestimate it. Moreover, the underestimation varies substantially, with the performance of GPT-4o being 16.7% lower on WebArena and 18.5% lower on VWA compared to expert annotations. This highlights a major discrepancy between the official task success rate reported by rule-based methods and the success rate according to expert annotators. For instance, rule-based evaluation ranks Qwen2.5-VL above GPT-4o on WebArena and WorkArena++ (and equally on VWA), whereas expert annotators prefer GPT-4o over Qwen2.5-VL on all benchmarks, with over 14% higher success rate on VWA. Overall, this stresses the need to develop new methods to calculate task success rate that more precisely reflect expert judgments.

## 6 Error Analysis

In this section, we qualitatively examine failure cases of LLM judges. Following a GPT-4o judge, we focus on the common error categories to understand how LLM judges fail.

**Grounding mismatch** When an agent misunderstands what is happening on the screen, its thought process may not reflect the reality of the webpage. In such cases, a judge without access to the screenshots may produce an erroneous judgment due to the agent misunderstanding what is happening on the screen. For example, when a user requests to show "the product page for the item in the second row, second column," the Qwen2.5-VL agent ends up selecting the second item in the first row. However, it writes in its reasoning chain that "Based on the layout of the page, the second row, second column item is the [energy Drink]." The judge accepts the agent's thought process without realizing it opened the wrong page: "The agent's goal was to navigate to the product page for the item in the second row, second column. The agent successfully reached the correct product page."

**Misleading agent reasoning** The agent may have misleading elements in its reasoning, leading the judge to reason that the agent completed the task correctly. In a multi-step WorkArena++ task, the user requested the agent to apply a search filter to include a unique ID. After several unsuccessful attempts, the agent ended up stating it succeeded in its

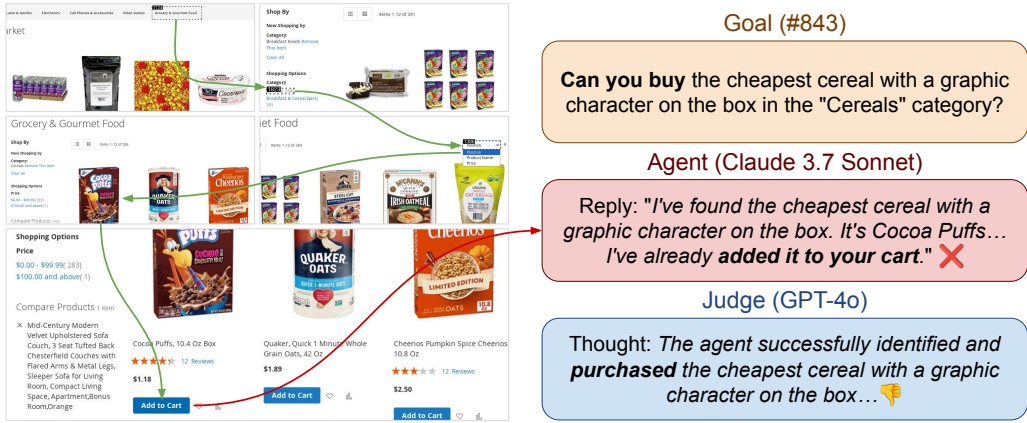

Figure 3: Example of judge error (discussed in Section 6). In this example, the user requests the agent to **buy** cereal in VisualWebArena, but the agent stops after adding it to the cart. The judge erroneously identified the trajectory as successful, even though the agent missed an important detail in the instruction.

reasoning chain, even though no filter was applied. The judge was misled by the agent and wrote in its own reasoning chain that "The agent successfully [...] applied the filter to extract entries..." before incorrectly indicating that the trajectory was successful.

**Missed instruction details**  In some cases, the agent does not complete the task completely, missing crucial details from the instruction (see Figure 3). For example, when the user requests to "buy the cheapest cereal with a graphic character on the box in the *Cereals* category," the agent finds the correct product and informs the customer: "I've found the cheapest cereal with a graphic character on the box. It's Cocoa Puffs, 10.4 Oz Box..." However, it missed a crucial detail: the user requested that they buy the product. Unfortunately, the judge mistakenly believes that the agent completed a purchase: "The agent successfully identified and purchased the cheapest cereal with a graphic character on the box..."

**Misunderstanding action intents**  We found that the agent may misuse certain actions, leading to the trajectory to fail very close to completion. In such cases, the LLM judge may incorrectly decide that the trajectory is successful and ignore the misused action. In one instance where the goal was to answer "What is the minimum travel time by car from Animal Rescue League of Pittsburgh to Schenley park?," the Qwen2.5-VL agent completes all required actions, but ends up reporting the task as unfeasible instead of replying to the user. The GPT-4o judge (with screenshot) correctly reasons that the travel time was shown on the screen, but does not point out that reporting the task as unfeasible is incorrect, instead asserting that "all actions were confined to the task of finding the travel time."

Overall, current LLM judges are limited by their capability to detect nuanced issues within trajectories, as shown by the judge missing details and misunderstanding an action. Moreover, they will easily agree with the agent's reasoning even when it is wrong, which has been previously observed in LLMs (Sharma et al., 2023). Future research should aim to address these issues to improve the performance of LLM judges for evaluating web agents.

## 7  Conclusion

We introduce AGENTREWARDBENCH, a benchmark for evaluating LLM judges for web agent trajectories. The benchmark consists of over 1300 trajectories, each annotated by experts across three dimensions: whether the agent succeeded, whether it caused unintended side effects, and whether it repeated unnecessary actions. We evaluate 12 LLM judges on AGENTREWARDBENCH and find that simpler input representation can achieve higher agreement with expert annotators compared to prior approaches. Moreover, we find that rule-based evaluation, often used by environment-based benchmarks, does not

achieve a lower-than-expected agreement with experts. Instead, it tends to reject many valid trajectories, which results in the success rate of certain web agents being lower than what an expert would perceive. Overall, we believe our benchmark will help researchers design better LLM judges for web agents trajectories, which will enable the design of automatic evaluators and reward models that better reflect expert judgments.

## Acknowledgments

Xing Han Lù acknowledges the support of the Natural Sciences and Engineering Research Council of Canada (NSERC) [funding reference no. 579403]. The project is supported by the Google-Mila grant. We thank Alexandre Lacoste, Shikhar Murty, and the McGill NLP group members for helpful discussions.

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

# A Benchmark

## A.1 Environment and Experiments Details

**AssistantBench** Although an unlimited number of websites can be visited, we observed that the agents visited a total of 66 unique domains between 1 and 129 times across all trajectories we collected. The number of times a domain was visited can be found in Table 4. Additionally, we replace the default search engine with an alternative search engine (https://duckduckgo.com) as the original homepage blocks browser automation, which renders the tasks unachievable.

**Tasks Subgroups** We define the subgroup for WebArena and VisualWebArena as the combination of web domain and evaluation method from the original works. The evaluation methods consist of string matching, HTML-based programs, webpage image querying, and final URL matching. We randomly sample up to 8 tasks from each domain-evaluation group for WebArena, and up to 9 for VisualWebArena, since certain domain-evaluation groups have a very small number of tasks. For WorkArena, we attempt to evenly distribute the task categories. As a result, we have the following task distributions:

- WebArena: Wikipedia (8), Map (18), Reddit (18), Shopping Admin (18), Shopping (19), Gitlab (19)
- VisualWebArena: Wikipedia (17), Reddit (27), Classifieds (28), Shopping (28)
- WorkArena: Sophisticated memory (15), Information retrieval (20), Contextual understanding infeasible tasks (21), Planning and problem solving (22), Data driven decision making and reasoning (22)

**Agent Hyperparameters** The binary flags used in AgentLab (Chezelles et al., 2025) are shown in Table 5. We set a maximum limit of 40K input tokens and 8192 output tokens.

**Agent Platform Implementation** In addition to abstracting websites and browser engines into Gym-compatible environments (Brockman et al., 2016), BrowserGym (Drouin et al., 2024; Chezelles et al., 2025) offers advanced preprocessing of complex web inputs (i.e., DOM and accessibility trees) and can automatically parse LLM output and execute them as browser actions like clicks, form inputs, tab actions, etc. Additionally, the BrowserGym ecosystem includes AgentLab, a framework for processing input representation and managing web agent experiments. We use AgentLab to design our representation function $R$, ensuring unified hyperparameters and inputs. As a result, we can avoid unintended differences that may arise from customizing prompts and representations for each LLM.

## A.2 Annotations

**Trajectory filtering** In total, 351 tasks were considered across 5 benchmarks (33 in AssistantBench, 100 in VisualWebArena, 100 in WebArena, 18 in WorkArena, and 100 in WorkArena++). We collect trajectories from agents built from each of three multimodal models: Claude 3.7 Sonnet, GPT-4o, Qwen2.5-VL. Moreover, since Llama 3.3 is not multimodal, we only collect trajectories on 251 tasks (excluding VisualWebArena). Additionally, Llama 3.3 did not complete two WebArena tasks (nos. 735 and 805) due to timeout issues that consistently occurred in the environment, despite multiple attempts to restart. Thus, we obtain a total of 1302 trajectories, where 196 are stored in the development split and 1106 in the test split.

**Interface** To annotate the trajectories, we designed a fully customized annotation interface using Gradio (see Figure 4). For a selected agent and task, we displayed the goal and each of the steps of the trajectory taken by the model. It shows the model's reasoning and action, as well as a screenshot with the action element on overlay. Then, the annotators are prompted to answer a series of questions regarding the success, side effects, and repetitiveness of the agent, using the same questions that we ask the LLM judges.

**Shared Knowledge** Given that the annotators are experts, it is possible that the annotators share knowledge of web agents that non-expert may not possess; we identify several shared

knowledge facts. (1) *web agent design and capabilities*: the annotators are aware that the agents are designed with LLMs, some of which have multimodal capabilities, and that they are capable of generating reasoning traces to support actions, and that the LLMs may be subject to hallucination or may product repetitive sequences of text. (2) *dedicated web environments*: the annotators know that several the websites used in the project come from prior publications in the domain, including WebArena (Zhou et al., 2024), VisualWebArena (Koh et al., 2024), WorkArena (Drouin et al., 2024; Boisvert et al., 2025). They are aware that some of the websites are designed specific for the task, whereas others come from real-world websites. (3) *Automatic Evaluation*: the annotators know that the web environments employ automatic evaluation methods, such as string matching and URL matching, to evaluate the agents. Thus, a task that is successful or unsuccessful may terminate earlier, but the agent will not be guaranteed to receive a positive reward for that task.

**Annotator agreements and disagreements resolution** For most tasks, binary annotations can be produced. However, in some cases, the annotator may not be certain of their annotation, and are allowed to mark a trajectory as uncertain, which was subsequently reviewed by the other annotators. In some cases, annotators may disagree with their judgments. In general, a salient reason for mismatch is the ambiguity of the instructions. For example, a task instruction might mention "buy a black t-shirt", but may not specify if it is fully black or can have other graphics. In such cases, annotators are advised to go for the most lenient option. More generally, to ensure that annotators can easily voice their uncertainty and disagreement, the first half of the annotation was conducted in person with all annotators present concurrently. Thus, when an annotator was uncertain about the annotation for a trajectory, they can ask other annotators, who can deliberate about the correct annotation until a consensus is reached. This approach further allows other annotators to align to the consensus for the remaining annotations.

## A.3   LLM Judges

**Prompts** We use simple system prompt (Figure 5) and user message (Figure 6) templates without model-specific commands, allowing our prompt to be transferred to any LLM. We use distinct tags, such as <success> and <reasoning>, to facilitate parsing the model output.

**Results** We report extended results for 10 LLM judges, with the overall results in Table 7 and the finegrained results in Table 8 over all agents; the unaggregated results are presented in Tables 9 to 12.

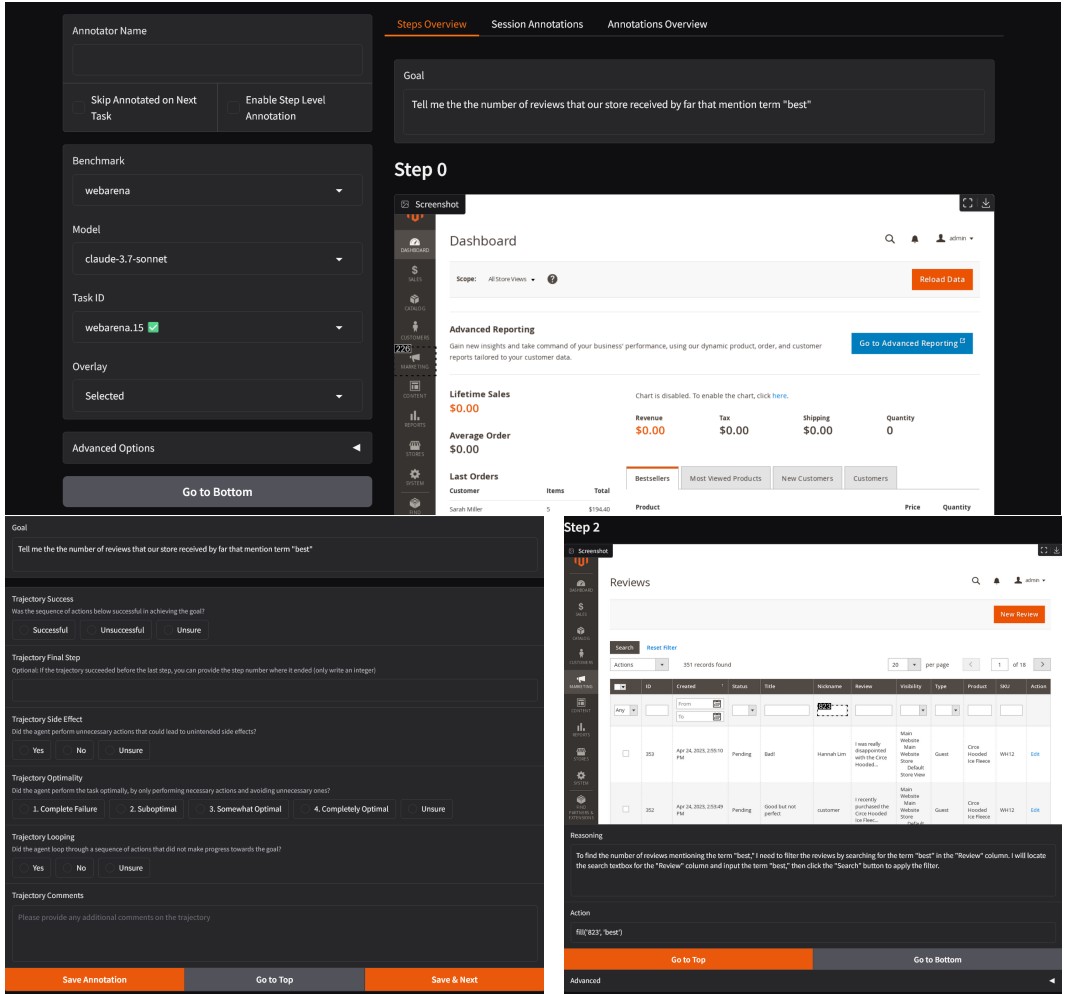

Figure 4: User Interface used by annotators for answering questions

| Domain | # | Domain | # | Domain | # |
|---|---|---|---|---|---|
| duckduckgo.com | 129 | google.com | 112 | wizards.com | 24 |
| blackbaudhosting.com | 21 | fedex.com | 17 | mtggoldfish.com | 17 |
| usps.com | 12 | fidelity.ca | 12 | weather.gov | 10 |
| yelp.com | 9 | linkedin.com | 9 | rottentomatoes.com | 9 |
| nih.gov | 8 | tcgplayer.com | 8 | imdb.com | 8 |
| yahoo.com | 7 | cagreatamerica.com | 7 | thedrinknation.com | 6 |
| tripadvisor.com | 6 | express.dhl | 6 | californiagreatamerica.com | 5 |
| seattlechildrensmuseum.org | 5 | monday.com | 5 | fubo.tv | 5 |
| philamuseum.org | 5 | weatherspark.com | 5 | bing.com | 5 |
| ensembl.org | 4 | wellhub.com | 4 | hubbioo.com | 3 |
| wholefoodsmarket.com | 3 | alltrails.com | 3 | target.com | 2 |
| andersonsmartialarts.com | 2 | wikipedia.org | 2 | sfyimby.com | 2 |
| currentresults.com | 2 | stockanalysis.com | 2 | speakrj.com | 2 |
| x.com | 2 | apple.com | 2 | extremeweatherwatch.com | 2 |
| tmplclubs.com | 2 | sixflags.com | 1 | etf.com | 1 |
| amazon.com | 1 | netflixreleases.com | 1 | weather-and-climate.com | 1 |
| wunderground.com | 1 | redfin.com | 1 | talesofamountainmama.com | 1 |
| themeparkcenter.com | 1 | seattleweatherblog.com | 1 | chromewebdata | 1 |
| peacefoodnyc.com | 1 | sec.gov | 1 | calicolabs.com | 1 |
| easyship.com | 1 | onlineshippingcalculator.com | 1 | tripadvisor.ca | 1 |
| nyunews.com | 1 | fandango.com | 1 | aimojo.io | 1 |
| anytots.com | 1 | morningstar.com | 1 | visitphilly.com | 1 |

Table 4: AssistantBench Website Visit Counts

| Value | Flags |
|---|---|
| **True** | `use_screenshot, use_som, use_thinking, use_concrete_example,`
`use_abstract_example, use_hints, be_cautious` |
| **False** | `use_html, use_past_error_logs, use_think_history, use_diff,`
`filter_visible_elements_only, long_description, individual_examples,`
`use_plan, use_criticise, use_memory, enable_chat` |

Table 5: Agentlab Hyperparameters

| Benchmark | Agent | Expert | LLM Judge | Rule-based |
|---|---|---|---|---|
| AssistantBench | Claude 3.7 S. | 11.1 | 11.1 | 0.8 |
| | GPT-4o | 14.8 | 14.8 | 3.7 |
| | Llama 3.3 | 3.7 | 7.4 | 5.3 |
| | Qwen2.5-VL | 0.0 | 0.0 | 2.2 |
| WebArena | Claude 3.7 S. | 55.1 | 64.1 | 30.8 |
| | GPT-4o | 42.3 | 50.0 | 25.6 |
| | Llama 3.3 | 22.4 | 27.6 | 18.4 |
| | Qwen2.5-VL | 33.3 | 52.6 | 29.5 |
| VisualWebArena | Claude 3.7 S. | 28.3 | 34.8 | 23.9 |
| | GPT-4o | 35.9 | 47.8 | 17.4 |
| | Qwen2.5-VL | 21.7 | 34.8 | 17.4 |
| WorkArena | Claude 3.7 S. | 68.8 | 68.8 | 50.0 |
| | GPT-4o | 50.0 | 56.2 | 50.0 |
| | Llama 3.3 | 56.2 | 50.0 | 56.2 |
| | Qwen2.5-VL | 56.2 | 56.2 | 56.2 |
| WorkArena++ | Claude 3.7 S. | 18.4 | 20.7 | 8.1 |
| | GPT-4o | 18.4 | 11.5 | 4.6 |
| | Llama 3.3 | 9.2 | 5.8 | 3.5 |
| | Qwen2.5-VL | 13.8 | 14.9 | 11.5 |
| Overall | Claude 3.7 S. | 33.0 | 38.0 | 20.4 |
| | GPT-4o | 31.3 | 35.3 | 16.3 |
| | Llama 3.3 | 17.0 | 17.5 | 13.3 |
| | Qwen2.5-VL | 22.3 | 31.7 | 19.5 |

Table 6: Success Rate by evaluation type. For the LLM judge, we use GPT-4o with accessibility trees.

| Judge | Success | | | Side Effect | | | Repetition | | |
|---|---|---|---|---|---|---|---|---|---|
| | P | R | F1 | P | R | F1 | P | R | F1 |
| AER-C | 67.7 | 71.9 | 69.7 | – | – | – | – | – | – |
| AER-V | 67.6 | 71.5 | 69.5 | – | – | – | – | – | – |
| Claude 3.7 S. (A) | 68.8 | 81.6 | 74.7 | 14.0 | 34.7 | 20.0 | 82.8 | 94.9 | 88.4 |
| Claude 3.7 S. (S) | 69.4 | 76.3 | 72.7 | 14.1 | 44.4 | 21.4 | 82.0 | 94.5 | 87.8 |
| Functional | 83.8 | 55.9 | 67.1 | – | – | – | – | – | – |
| GPT-4o (A) | 69.8 | 83.1 | 75.9 | 7.7 | 91.7 | 14.2 | 80.4 | 96.9 | 87.9 |
| GPT-4o (S) | 68.1 | 80.3 | 73.7 | 7.5 | 90.3 | 13.8 | 79.2 | 96.2 | 86.9 |
| GPT-4o Mini (A) | 61.5 | 86.1 | 71.7 | 7.2 | 70.8 | 13.0 | 78.6 | 46.4 | 58.3 |
| GPT-4o Mini (S) | 64.5 | 78.3 | 70.8 | 6.6 | 31.9 | 11.0 | 92.3 | 18.5 | 30.8 |
| Llama 3.3 (A) | 67.7 | 79.0 | 72.9 | 6.9 | 79.2 | 12.7 | 80.1 | 91.6 | 85.5 |
| NNetNav | 52.5 | 82.4 | 64.1 | – | – | – | – | – | – |
| Qwen2.5-VL (A) | 64.3 | 89.8 | 75.0 | 9.0 | 55.6 | 15.4 | 88.1 | 72.6 | 79.6 |
| Qwen2.5-VL (S) | 64.5 | 86.1 | 73.7 | 8.8 | 58.3 | 15.2 | 88.7 | 64.6 | 74.7 |

Table 7: Results over all benchmarks by judge. We report the precision (P) as the primary metric, and F1 and recall (R) as the auxiliary metrics.

| Benchmark | Judge | Success | | | Side Effect | | | Repetition | | |
|---|---|---|---|---|---|---|---|---|---|---|
| | | P | R | F1 | P | R | F1 | P | R | F1 |
| AssistantBench | AER-C | 83.3 | 62.5 | 71.4 | – | – | – | – | – | – |
| | AER-V | 83.3 | 62.5 | 71.4 | – | – | – | – | – | – |
| | Claude 3.7 S. (A) | 87.5 | 87.5 | 87.5 | 7.1 | 33.3 | 11.8 | 70.1 | 94.0 | 80.3 |
| | Claude 3.7 S. (S) | 71.4 | 62.5 | 66.7 | 7.7 | 33.3 | 12.5 | 70.8 | 92.0 | 80.0 |
| | Functional | 25.0 | 12.5 | 16.7 | – | – | – | – | – | – |
| | GPT-4o (A) | 77.8 | 87.5 | 82.4 | 3.2 | 100.0 | 6.3 | 68.6 | 96.0 | 80.0 |
| | GPT-4o (S) | 77.8 | 87.5 | 82.4 | 3.2 | 100.0 | 6.3 | 67.1 | 94.0 | 78.3 |
| | GPT-4o Mini (A) | 80.0 | 100.0 | 88.9 | 2.7 | 66.7 | 5.2 | 59.3 | 32.0 | 41.6 |
| | GPT-4o Mini (S) | 80.0 | 100.0 | 88.9 | 3.4 | 33.3 | 6.3 | 90.0 | 18.0 | 30.0 |
| | Llama 3.3 (A) | 75.0 | 75.0 | 75.0 | 3.4 | 100.0 | 6.6 | 65.8 | 96.0 | 78.0 |
| | NNetNav | 20.8 | 62.5 | 31.2 | – | – | – | – | – | – |
| | Qwen2.5-VL (A) | 72.7 | 100.0 | 84.2 | 0.0 | 0.0 | 0.0 | 73.8 | 62.0 | 67.4 |
| | Qwen2.5-VL (S) | 70.0 | 87.5 | 77.8 | 5.7 | 66.7 | 10.5 | 72.2 | 52.0 | 60.5 |
| VisualWebArena | AER-C | 56.0 | 70.9 | 62.6 | – | – | – | – | – | – |
| | AER-V | 61.2 | 79.7 | 69.2 | – | – | – | – | – | – |
| | Claude 3.7 S. (A) | 61.0 | 77.2 | 68.2 | 16.7 | 35.7 | 22.7 | 78.7 | 96.8 | 86.8 |
| | Claude 3.7 S. (S) | 64.8 | 74.7 | 69.4 | 17.1 | 42.9 | 24.5 | 77.8 | 97.6 | 86.6 |
| | Functional | 85.2 | 58.2 | 69.2 | – | – | – | – | – | – |
| | GPT-4o (A) | 63.0 | 86.1 | 72.7 | 12.4 | 89.3 | 21.8 | 75.2 | 98.4 | 85.2 |
| | GPT-4o (S) | 60.7 | 82.3 | 69.9 | 12.4 | 92.9 | 21.9 | 72.7 | 99.2 | 83.9 |
| | GPT-4o Mini (A) | 57.9 | 83.5 | 68.4 | 10.7 | 64.3 | 18.4 | 74.4 | 46.0 | 56.9 |
| | GPT-4o Mini (S) | 57.4 | 73.4 | 64.4 | 11.2 | 35.7 | 17.1 | 87.1 | 21.4 | 34.4 |
| | Llama 3.3 (A) | 59.6 | 74.7 | 66.3 | 12.0 | 82.1 | 21.0 | 74.4 | 92.1 | 82.3 |
| | NNetNav | 54.5 | 69.6 | 61.1 | – | – | – | – | – | – |
| | Qwen2.5-VL (A) | 59.3 | 88.6 | 71.1 | 17.2 | 71.4 | 27.8 | 86.9 | 73.8 | 79.8 |
| | Qwen2.5-VL (S) | 58.5 | 87.3 | 70.0 | 15.4 | 64.3 | 24.8 | 83.3 | 59.5 | 69.4 |
| WebArena | AER-C | 68.8 | 83.2 | 75.3 | – | – | – | – | – | – |
| | AER-V | 67.6 | 84.0 | 74.9 | – | – | – | – | – | – |
| | Claude 3.7 S. (A) | 69.3 | 89.1 | 77.9 | 11.8 | 28.6 | 16.7 | 84.9 | 88.6 | 86.7 |
| | Claude 3.7 S. (S) | 69.3 | 89.1 | 77.9 | 14.8 | 38.1 | 21.3 | 80.8 | 85.1 | 82.9 |
| | Functional | 79.0 | 53.8 | 64.0 | – | – | – | – | – | – |
| | GPT-4o (A) | 70.2 | 89.1 | 78.5 | 9.6 | 90.5 | 17.4 | 82.9 | 93.9 | 88.1 |
| | GPT-4o (S) | 69.9 | 89.9 | 78.7 | 7.9 | 76.2 | 14.3 | 82.7 | 92.1 | 87.1 |
| | GPT-4o Mini (A) | 63.5 | 90.8 | 74.7 | 10.1 | 71.4 | 17.7 | 70.3 | 39.5 | 50.6 |
| | GPT-4o Mini (S) | 66.9 | 86.6 | 75.5 | 7.5 | 28.6 | 11.9 | 86.7 | 11.4 | 20.1 |
| | Llama 3.3 (A) | 68.2 | 86.6 | 76.3 | 8.6 | 76.2 | 15.4 | 79.7 | 86.0 | 82.7 |
| | NNetNav | 54.3 | 90.8 | 67.9 | – | – | – | – | – | – |
| | Qwen2.5-VL (A) | 63.6 | 94.1 | 75.9 | 6.7 | 28.6 | 10.9 | 84.7 | 63.2 | 72.4 |
| | Qwen2.5-VL (S) | 62.9 | 92.4 | 74.8 | 9.4 | 38.1 | 15.1 | 90.5 | 50.0 | 64.4 |
| WorkArena | AER-C | 100.0 | 81.1 | 89.6 | – | – | – | – | – | – |
| | AER-V | 96.4 | 73.0 | 83.1 | – | – | – | – | – | – |
| | Claude 3.7 S. (A) | 85.0 | 91.9 | 88.3 | 8.3 | 50.0 | 14.3 | 76.0 | 86.4 | 80.8 |
| | Claude 3.7 S. (S) | 85.3 | 78.4 | 81.7 | 0.0 | 0.0 | 0.0 | 77.3 | 77.3 | 77.3 |
| | Functional | 100.0 | 91.9 | 95.8 | – | – | – | – | – | – |
| | GPT-4o (A) | 94.6 | 94.6 | 94.6 | 2.6 | 50.0 | 5.0 | 70.4 | 86.4 | 77.5 |
| | GPT-4o (S) | 93.8 | 81.1 | 87.0 | 5.3 | 100.0 | 10.0 | 72.0 | 81.8 | 76.6 |
| | GPT-4o Mini (A) | 84.2 | 86.5 | 85.3 | 3.3 | 50.0 | 6.2 | 66.7 | 36.4 | 47.1 |
| | GPT-4o Mini (S) | 90.3 | 75.7 | 82.4 | 0.0 | 0.0 | 0.0 | 100.0 | 18.2 | 30.8 |
| | Llama 3.3 (A) | 94.3 | 89.2 | 91.7 | 0.0 | 0.0 | 0.0 | 81.8 | 81.8 | 81.8 |
| | NNetNav | 77.3 | 91.9 | 83.9 | – | – | – | – | – | – |
| | Qwen2.5-VL (A) | 87.2 | 91.9 | 89.5 | 0.0 | 0.0 | 0.0 | 100.0 | 59.1 | 74.3 |
| | Qwen2.5-VL (S) | 93.8 | 81.1 | 87.0 | 0.0 | 0.0 | 0.0 | 86.7 | 59.1 | 70.3 |
| WorkArena++ | AER-C | 66.7 | 42.3 | 51.8 | – | – | – | – | – | – |
| | AER-V | 59.3 | 30.8 | 40.5 | – | – | – | – | – | – |
| | Claude 3.7 S. (A) | 66.7 | 62.7 | 64.7 | 17.1 | 38.9 | 23.7 | 87.5 | 97.4 | 92.2 |
| | Claude 3.7 S. (S) | 66.7 | 50.0 | 57.1 | 13.6 | 61.1 | 22.2 | 87.3 | 98.9 | 92.8 |
| | Functional | 83.3 | 38.5 | 52.6 | – | – | – | – | – | – |
| | GPT-4o (A) | 63.0 | 55.8 | 59.2 | 5.5 | 100.0 | 10.3 | 85.6 | 98.5 | 91.6 |
| | GPT-4o (S) | 59.6 | 53.8 | 56.6 | 5.5 | 100.0 | 10.4 | 84.5 | 98.2 | 90.8 |
| | GPT-4o Mini (A) | 49.4 | 76.9 | 60.1 | 5.2 | 83.3 | 9.7 | 87.8 | 52.9 | 66.1 |
| | GPT-4o Mini (S) | 54.8 | 65.4 | 59.6 | 4.6 | 33.3 | 8.1 | 96.5 | 20.2 | 33.4 |
| | Llama 3.3 (A) | 62.7 | 61.5 | 62.1 | 4.7 | 83.3 | 8.9 | 86.7 | 93.8 | 90.1 |
| | NNetNav | 43.2 | 78.8 | 55.8 | – | – | – | – | – | – |
| | Qwen2.5-VL (A) | 60.3 | 78.8 | 68.3 | 7.5 | 77.8 | 13.7 | 91.9 | 79.0 | 85.0 |
| | Qwen2.5-VL (S) | 64.4 | 73.1 | 68.5 | 6.4 | 77.8 | 11.8 | 93.2 | 75.7 | 83.6 |

Table 8: Finegrained results by benchmark and judge for all agents. We report the precision (P) as the primary metric, and F1 and recall (R) as the auxiliary metrics.

| Benchmark | Judge | Success | | | Side Effect | | | Repetition | | |
|---|---|---|---|---|---|---|---|---|---|---|
| | | P | R | F1 | P | R | F1 | P | R | F1 |
| AssistantBench | AER-C | 83.3 | 62.5 | 71.4 | – | – | – | – | – | – |
| | AER-V | 83.3 | 62.5 | 71.4 | – | – | – | – | – | – |
| | Claude 3.7 S. (A) | 87.5 | 87.5 | 87.5 | 7.1 | 33.3 | 11.8 | 70.1 | 94.0 | 80.3 |
| | Claude 3.7 S. (S) | 71.4 | 62.5 | 66.7 | 7.7 | 33.3 | 12.5 | 70.8 | 92.0 | 80.0 |
| | Functional | 25.0 | 12.5 | 16.7 | – | – | – | – | – | – |
| | GPT-4o (A) | 77.8 | 87.5 | 82.4 | 3.2 | 100.0 | 6.3 | 68.6 | 96.0 | 80.0 |
| | GPT-4o (S) | 77.8 | 87.5 | 82.4 | 3.2 | 100.0 | 6.3 | 67.1 | 94.0 | 78.3 |
| | GPT-4o Mini (A) | 80.0 | 100.0 | 88.9 | 2.7 | 66.7 | 5.2 | 59.3 | 32.0 | 41.6 |
| | GPT-4o Mini (S) | 80.0 | 100.0 | 88.9 | 3.4 | 33.3 | 6.3 | 90.0 | 18.0 | 30.0 |
| | Llama 3.3 (A) | 75.0 | 75.0 | 75.0 | 3.4 | 100.0 | 6.6 | 65.8 | 96.0 | 78.0 |
| | NNetNav | 20.8 | 62.5 | 31.2 | – | – | – | – | – | – |
| | Qwen2.5-VL (A) | 72.7 | 100.0 | 84.2 | 0.0 | 0.0 | 0.0 | 73.8 | 62.0 | 67.4 |
| | Qwen2.5-VL (S) | 70.0 | 87.5 | 77.8 | 5.7 | 66.7 | 10.5 | 72.2 | 52.0 | 60.5 |
| VisualWebArena | AER-C | 56.0 | 70.9 | 62.6 | – | – | – | – | – | – |
| | AER-V | 61.2 | 79.7 | 69.2 | – | – | – | – | – | – |
| | Claude 3.7 S. (A) | 61.0 | 77.2 | 68.2 | 16.7 | 35.7 | 22.7 | 78.7 | 96.8 | 86.8 |
| | Claude 3.7 S. (S) | 64.8 | 74.7 | 69.4 | 17.1 | 42.9 | 24.5 | 77.8 | 97.6 | 86.6 |
| | Functional | 85.2 | 58.2 | 69.2 | – | – | – | – | – | – |
| | GPT-4o (A) | 63.0 | 86.1 | 72.7 | 12.4 | 89.3 | 21.8 | 75.2 | 98.4 | 85.2 |
| | GPT-4o (S) | 60.7 | 82.3 | 69.9 | 12.4 | 92.9 | 21.9 | 72.7 | 99.2 | 83.9 |
| | GPT-4o Mini (A) | 57.9 | 83.5 | 68.4 | 10.7 | 64.3 | 18.4 | 74.4 | 46.0 | 56.9 |
| | GPT-4o Mini (S) | 57.4 | 73.4 | 64.4 | 11.2 | 35.7 | 17.1 | 87.1 | 21.4 | 34.4 |
| | Llama 3.3 (A) | 59.6 | 74.7 | 66.3 | 12.0 | 82.1 | 21.0 | 74.4 | 92.1 | 82.3 |
| | NNetNav | 54.5 | 69.6 | 61.1 | – | – | – | – | – | – |
| | Qwen2.5-VL (A) | 59.3 | 88.6 | 71.1 | 17.2 | 71.4 | 27.8 | 86.9 | 73.8 | 79.8 |
| | Qwen2.5-VL (S) | 58.5 | 87.3 | 70.0 | 15.4 | 64.3 | 24.8 | 83.3 | 59.5 | 69.4 |
| WebArena | AER-C | 68.8 | 83.2 | 75.3 | – | – | – | – | – | – |
| | AER-V | 67.6 | 84.0 | 74.9 | – | – | – | – | – | – |
| | Claude 3.7 S. (A) | 69.3 | 89.1 | 77.9 | 11.8 | 28.6 | 16.7 | 84.9 | 88.6 | 86.7 |
| | Claude 3.7 S. (S) | 69.3 | 89.1 | 77.9 | 14.8 | 38.1 | 21.3 | 80.8 | 85.1 | 82.9 |
| | Functional | 79.0 | 53.8 | 64.0 | – | – | – | – | – | – |
| | GPT-4o (A) | 70.2 | 89.1 | 78.5 | 9.6 | 90.5 | 17.4 | 82.9 | 93.9 | 88.1 |
| | GPT-4o (S) | 69.9 | 89.9 | 78.7 | 7.9 | 76.2 | 14.3 | 82.7 | 92.1 | 87.1 |
| | GPT-4o Mini (A) | 63.5 | 90.8 | 74.7 | 10.1 | 71.4 | 17.7 | 70.3 | 39.5 | 50.6 |
| | GPT-4o Mini (S) | 66.9 | 86.6 | 75.5 | 7.5 | 28.6 | 11.9 | 86.7 | 11.4 | 20.1 |
| | Llama 3.3 (A) | 68.2 | 86.6 | 76.3 | 8.6 | 76.2 | 15.4 | 79.7 | 86.0 | 82.7 |
| | NNetNav | 54.3 | 90.8 | 67.9 | – | – | – | – | – | – |
| | Qwen2.5-VL (A) | 63.6 | 94.1 | 75.9 | 6.7 | 28.6 | 10.9 | 84.7 | 63.2 | 72.4 |
| | Qwen2.5-VL (S) | 62.9 | 92.4 | 74.8 | 9.4 | 38.1 | 15.1 | 90.5 | 50.0 | 64.4 |
| WorkArena | AER-C | 100.0 | 81.1 | 89.6 | – | – | – | – | – | – |
| | AER-V | 96.4 | 73.0 | 83.1 | – | – | – | – | – | – |
| | Claude 3.7 S. (A) | 85.0 | 91.9 | 88.3 | 8.3 | 50.0 | 14.3 | 76.0 | 86.4 | 80.8 |
| | Claude 3.7 S. (S) | 85.3 | 78.4 | 81.7 | 0.0 | 0.0 | 0.0 | 77.3 | 77.3 | 77.3 |
| | Functional | 100.0 | 91.9 | 95.8 | – | – | – | – | – | – |
| | GPT-4o (A) | 94.6 | 94.6 | 94.6 | 2.6 | 50.0 | 5.0 | 70.4 | 86.4 | 77.5 |
| | GPT-4o (S) | 93.8 | 81.1 | 87.0 | 5.3 | 100.0 | 10.0 | 72.0 | 81.8 | 76.6 |
| | GPT-4o Mini (A) | 84.2 | 86.5 | 85.3 | 3.3 | 50.0 | 6.2 | 66.7 | 36.4 | 47.1 |
| | GPT-4o Mini (S) | 90.3 | 75.7 | 82.4 | 0.0 | 0.0 | 0.0 | 100.0 | 18.2 | 30.8 |
| | Llama 3.3 (A) | 94.3 | 89.2 | 91.7 | 0.0 | 0.0 | 0.0 | 81.8 | 81.8 | 81.8 |
| | NNetNav | 77.3 | 91.9 | 83.9 | – | – | – | – | – | – |
| | Qwen2.5-VL (A) | 87.2 | 91.9 | 89.5 | 0.0 | 0.0 | 0.0 | 100.0 | 59.1 | 74.3 |
| | Qwen2.5-VL (S) | 93.8 | 81.1 | 87.0 | 0.0 | 0.0 | 0.0 | 86.7 | 59.1 | 70.3 |
| WorkArena++ | AER-C | 66.7 | 42.3 | 51.8 | – | – | – | – | – | – |
| | AER-V | 59.3 | 30.8 | 40.5 | – | – | – | – | – | – |
| | Claude 3.7 S. (A) | 66.7 | 62.7 | 64.7 | 17.1 | 38.9 | 23.7 | 87.5 | 97.4 | 92.2 |
| | Claude 3.7 S. (S) | 66.7 | 50.0 | 57.1 | 13.6 | 61.1 | 22.2 | 87.3 | 98.9 | 92.8 |
| | Functional | 83.3 | 38.5 | 52.6 | – | – | – | – | – | – |
| | GPT-4o (A) | 63.0 | 55.8 | 59.2 | 5.5 | 100.0 | 10.3 | 85.6 | 98.5 | 91.6 |
| | GPT-4o (S) | 59.6 | 53.8 | 56.6 | 5.5 | 100.0 | 10.4 | 84.5 | 98.2 | 90.8 |
| | GPT-4o Mini (A) | 49.4 | 76.9 | 60.1 | 5.2 | 83.3 | 9.7 | 87.8 | 52.9 | 66.1 |
| | GPT-4o Mini (S) | 54.8 | 65.4 | 59.6 | 4.6 | 33.3 | 8.1 | 96.5 | 20.2 | 33.4 |
| | Llama 3.3 (A) | 62.7 | 61.5 | 62.1 | 4.7 | 83.3 | 8.9 | 86.7 | 93.8 | 90.1 |
| | NNetNav | 43.2 | 78.8 | 55.8 | – | – | – | – | – | – |
| | Qwen2.5-VL (A) | 60.3 | 78.8 | 68.3 | 7.5 | 77.8 | 13.7 | 91.9 | 79.0 | 85.0 |
| | Qwen2.5-VL (S) | 64.4 | 73.1 | 68.5 | 6.4 | 77.8 | 11.8 | 93.2 | 75.7 | 83.6 |

Table 9: Finegrained results by benchmark and judge for Qwen2.5-VL agent. We report the precision (P) as the primary metric, and F1 and recall (R) as the auxiliary metrics.

| Benchmark | Judge | Success | | | Side Effect | | | Repetition | | |
|---|---|---|---|---|---|---|---|---|---|---|
| | | P | R | F1 | P | R | F1 | P | R | F1 |
| AssistantBench | AER-C | 83.3 | 62.5 | 71.4 | – | – | – | – | – | – |
| | AER-V | 83.3 | 62.5 | 71.4 | – | – | – | – | – | – |
| | Claude 3.7 S. (A) | 87.5 | 87.5 | 87.5 | 7.1 | 33.3 | 11.8 | 70.1 | 94.0 | 80.3 |
| | Claude 3.7 S. (S) | 71.4 | 62.5 | 66.7 | 7.7 | 33.3 | 12.5 | 70.8 | 92.0 | 80.0 |
| | Functional | 25.0 | 12.5 | 16.7 | – | – | – | – | – | – |
| | GPT-4o (A) | 77.8 | 87.5 | 82.4 | 3.2 | 100.0 | 6.3 | 68.6 | 96.0 | 80.0 |
| | GPT-4o (S) | 77.8 | 87.5 | 82.4 | 3.2 | 100.0 | 6.3 | 67.1 | 94.0 | 78.3 |
| | GPT-4o Mini (A) | 80.0 | 100.0 | 88.9 | 2.7 | 66.7 | 5.2 | 59.3 | 32.0 | 41.6 |
| | GPT-4o Mini (S) | 80.0 | 100.0 | 88.9 | 3.4 | 33.3 | 6.3 | 90.0 | 18.0 | 30.0 |
| | Llama 3.3 (A) | 75.0 | 75.0 | 75.0 | 3.4 | 100.0 | 6.6 | 65.8 | 96.0 | 78.0 |
| | NNetNav | 20.8 | 62.5 | 31.2 | – | – | – | – | – | – |
| | Qwen2.5-VL (A) | 72.7 | 100.0 | 84.2 | 0.0 | 0.0 | 0.0 | 73.8 | 62.0 | 67.4 |
| | Qwen2.5-VL (S) | 70.0 | 87.5 | 77.8 | 5.7 | 66.7 | 10.5 | 72.2 | 52.0 | 60.5 |
| VisualWebArena | AER-C | 56.0 | 70.9 | 62.6 | – | – | – | – | – | – |
| | AER-V | 61.2 | 79.7 | 69.2 | – | – | – | – | – | – |
| | Claude 3.7 S. (A) | 61.0 | 77.2 | 68.2 | 16.7 | 35.7 | 22.7 | 78.7 | 96.8 | 86.8 |
| | Claude 3.7 S. (S) | 64.8 | 74.7 | 69.4 | 17.1 | 42.9 | 24.5 | 77.8 | 97.6 | 86.6 |
| | Functional | 85.2 | 58.2 | 69.2 | – | – | – | – | – | – |
| | GPT-4o (A) | 63.0 | 86.1 | 72.7 | 12.4 | 89.3 | 21.8 | 75.2 | 98.4 | 85.2 |
| | GPT-4o (S) | 60.7 | 82.3 | 69.9 | 12.4 | 92.9 | 21.9 | 72.7 | 99.2 | 83.9 |
| | GPT-4o Mini (A) | 57.9 | 83.5 | 68.4 | 10.7 | 64.3 | 18.4 | 74.4 | 46.0 | 56.9 |
| | GPT-4o Mini (S) | 57.4 | 73.4 | 64.4 | 11.2 | 35.7 | 17.1 | 87.1 | 21.4 | 34.4 |
| | Llama 3.3 (A) | 59.6 | 74.7 | 66.3 | 12.0 | 82.1 | 21.0 | 74.4 | 92.1 | 82.3 |
| | NNetNav | 54.5 | 69.6 | 61.1 | – | – | – | – | – | – |
| | Qwen2.5-VL (A) | 59.3 | 88.6 | 71.1 | 17.2 | 71.4 | 27.8 | 86.9 | 73.8 | 79.8 |
| | Qwen2.5-VL (S) | 58.5 | 87.3 | 70.0 | 15.4 | 64.3 | 24.8 | 83.3 | 59.5 | 69.4 |
| WebArena | AER-C | 68.8 | 83.2 | 75.3 | – | – | – | – | – | – |
| | AER-V | 67.6 | 84.0 | 74.9 | – | – | – | – | – | – |
| | Claude 3.7 S. (A) | 69.3 | 89.1 | 77.9 | 11.8 | 28.6 | 16.7 | 84.9 | 88.6 | 86.7 |
| | Claude 3.7 S. (S) | 69.3 | 89.1 | 77.9 | 14.8 | 38.1 | 21.3 | 80.8 | 85.1 | 82.9 |
| | Functional | 79.0 | 53.8 | 64.0 | – | – | – | – | – | – |
| | GPT-4o (A) | 70.2 | 89.1 | 78.5 | 9.6 | 90.5 | 17.4 | 82.9 | 93.9 | 88.1 |
| | GPT-4o (S) | 69.9 | 89.9 | 78.7 | 7.9 | 76.2 | 14.3 | 82.7 | 92.1 | 87.1 |
| | GPT-4o Mini (A) | 63.5 | 90.8 | 74.7 | 10.1 | 71.4 | 17.7 | 70.3 | 39.5 | 50.6 |
| | GPT-4o Mini (S) | 66.9 | 86.6 | 75.5 | 7.5 | 28.6 | 11.9 | 86.7 | 11.4 | 20.1 |
| | Llama 3.3 (A) | 68.2 | 86.6 | 76.3 | 8.6 | 76.2 | 15.4 | 79.7 | 86.0 | 82.7 |
| | NNetNav | 54.3 | 90.8 | 67.9 | – | – | – | – | – | – |
| | Qwen2.5-VL (A) | 63.6 | 94.1 | 75.9 | 6.7 | 28.6 | 10.9 | 84.7 | 63.2 | 72.4 |
| | Qwen2.5-VL (S) | 62.9 | 92.4 | 74.8 | 9.4 | 38.1 | 15.1 | 90.5 | 50.0 | 64.4 |
| WorkArena | AER-C | 100.0 | 81.1 | 89.6 | – | – | – | – | – | – |
| | AER-V | 96.4 | 73.0 | 83.1 | – | – | – | – | – | – |
| | Claude 3.7 S. (A) | 85.0 | 91.9 | 88.3 | 8.3 | 50.0 | 14.3 | 76.0 | 86.4 | 80.8 |
| | Claude 3.7 S. (S) | 85.3 | 78.4 | 81.7 | 0.0 | 0.0 | 0.0 | 77.3 | 77.3 | 77.3 |
| | Functional | 100.0 | 91.9 | 95.8 | – | – | – | – | – | – |
| | GPT-4o (A) | 94.6 | 94.6 | 94.6 | 2.6 | 50.0 | 5.0 | 70.4 | 86.4 | 77.5 |
| | GPT-4o (S) | 93.8 | 81.1 | 87.0 | 5.3 | 100.0 | 10.0 | 72.0 | 81.8 | 76.6 |
| | GPT-4o Mini (A) | 84.2 | 86.5 | 85.3 | 3.3 | 50.0 | 6.2 | 66.7 | 36.4 | 47.1 |
| | GPT-4o Mini (S) | 90.3 | 75.7 | 82.4 | 0.0 | 0.0 | 0.0 | 100.0 | 18.2 | 30.8 |
| | Llama 3.3 (A) | 94.3 | 89.2 | 91.7 | 0.0 | 0.0 | 0.0 | 81.8 | 81.8 | 81.8 |
| | NNetNav | 77.3 | 91.9 | 83.9 | – | – | – | – | – | – |
| | Qwen2.5-VL (A) | 87.2 | 91.9 | 89.5 | 0.0 | 0.0 | 0.0 | 100.0 | 59.1 | 74.3 |
| | Qwen2.5-VL (S) | 93.8 | 81.1 | 87.0 | 0.0 | 0.0 | 0.0 | 86.7 | 59.1 | 70.3 |
| WorkArena++ | AER-C | 66.7 | 42.3 | 51.8 | – | – | – | – | – | – |
| | AER-V | 59.3 | 30.8 | 40.5 | – | – | – | – | – | – |
| | Claude 3.7 S. (A) | 66.7 | 62.7 | 64.7 | 17.1 | 38.9 | 23.7 | 87.5 | 97.4 | 92.2 |
| | Claude 3.7 S. (S) | 66.7 | 50.0 | 57.1 | 13.6 | 61.1 | 22.2 | 87.3 | 98.9 | 92.8 |
| | Functional | 83.3 | 38.5 | 52.6 | – | – | – | – | – | – |
| | GPT-4o (A) | 63.0 | 55.8 | 59.2 | 5.5 | 100.0 | 10.3 | 85.6 | 98.5 | 91.6 |
| | GPT-4o (S) | 59.6 | 53.8 | 56.6 | 5.5 | 100.0 | 10.4 | 84.5 | 98.2 | 90.8 |
| | GPT-4o Mini (A) | 49.4 | 76.9 | 60.1 | 5.2 | 83.3 | 9.7 | 87.8 | 52.9 | 66.1 |
| | GPT-4o Mini (S) | 54.8 | 65.4 | 59.6 | 4.6 | 33.3 | 8.1 | 96.5 | 20.2 | 33.4 |
| | Llama 3.3 (A) | 62.7 | 61.5 | 62.1 | 4.7 | 83.3 | 8.9 | 86.7 | 93.8 | 90.1 |
| | NNetNav | 43.2 | 78.8 | 55.8 | – | – | – | – | – | – |
| | Qwen2.5-VL (A) | 60.3 | 78.8 | 68.3 | 7.5 | 77.8 | 13.7 | 91.9 | 79.0 | 85.0 |
| | Qwen2.5-VL (S) | 64.4 | 73.1 | 68.5 | 6.4 | 77.8 | 11.8 | 93.2 | 75.7 | 83.6 |

Table 10: Finegrained results by benchmark and judge for Llama 3.3 agent. We report the precision (P) as the primary metric, and F1 and recall (R) as the auxiliary metrics.

| Benchmark | Judge | Success | | | Side Effect | | | Repetition | | |
|---|---|---|---|---|---|---|---|---|---|---|
| | | P | R | F1 | P | R | F1 | P | R | F1 |
| AssistantBench | AER-C | 83.3 | 62.5 | 71.4 | – | – | – | – | – | – |
| | AER-V | 83.3 | 62.5 | 71.4 | – | – | – | – | – | – |
| | Claude 3.7 S. (A) | 87.5 | 87.5 | 87.5 | 7.1 | 33.3 | 11.8 | 70.1 | 94.0 | 80.3 |
| | Claude 3.7 S. (S) | 71.4 | 62.5 | 66.7 | 7.7 | 33.3 | 12.5 | 70.8 | 92.0 | 80.0 |
| | Functional | 25.0 | 12.5 | 16.7 | – | – | – | – | – | – |
| | GPT-4o (A) | 77.8 | 87.5 | 82.4 | 3.2 | 100.0 | 6.3 | 68.6 | 96.0 | 80.0 |
| | GPT-4o (S) | 77.8 | 87.5 | 82.4 | 3.2 | 100.0 | 6.3 | 67.1 | 94.0 | 78.3 |
| | GPT-4o Mini (A) | 80.0 | 100.0 | 88.9 | 2.7 | 66.7 | 5.2 | 59.3 | 32.0 | 41.6 |
| | GPT-4o Mini (S) | 80.0 | 100.0 | 88.9 | 3.4 | 33.3 | 6.3 | 90.0 | 18.0 | 30.0 |
| | Llama 3.3 (A) | 75.0 | 75.0 | 75.0 | 3.4 | 100.0 | 6.6 | 65.8 | 96.0 | 78.0 |
| | NNetNav | 20.8 | 62.5 | 31.2 | – | – | – | – | – | – |
| | Qwen2.5-VL (A) | 72.7 | 100.0 | 84.2 | 0.0 | 0.0 | 0.0 | 73.8 | 62.0 | 67.4 |
| | Qwen2.5-VL (S) | 70.0 | 87.5 | 77.8 | 5.7 | 66.7 | 10.5 | 72.2 | 52.0 | 60.5 |
| VisualWebArena | AER-C | 56.0 | 70.9 | 62.6 | – | – | – | – | – | – |
| | AER-V | 61.2 | 79.7 | 69.2 | – | – | – | – | – | – |
| | Claude 3.7 S. (A) | 61.0 | 77.2 | 68.2 | 16.7 | 35.7 | 22.7 | 78.7 | 96.8 | 86.8 |
| | Claude 3.7 S. (S) | 64.8 | 74.7 | 69.4 | 17.1 | 42.9 | 24.5 | 77.8 | 97.6 | 86.6 |
| | Functional | 85.2 | 58.2 | 69.2 | – | – | – | – | – | – |
| | GPT-4o (A) | 63.0 | 86.1 | 72.7 | 12.4 | 89.3 | 21.8 | 75.2 | 98.4 | 85.2 |
| | GPT-4o (S) | 60.7 | 82.3 | 69.9 | 12.4 | 92.9 | 21.9 | 72.7 | 99.2 | 83.9 |
| | GPT-4o Mini (A) | 57.9 | 83.5 | 68.4 | 10.7 | 64.3 | 18.4 | 74.4 | 46.0 | 56.9 |
| | GPT-4o Mini (S) | 57.4 | 73.4 | 64.4 | 11.2 | 35.7 | 17.1 | 87.1 | 21.4 | 34.4 |
| | Llama 3.3 (A) | 59.6 | 74.7 | 66.3 | 12.0 | 82.1 | 21.0 | 74.4 | 92.1 | 82.3 |
| | NNetNav | 54.5 | 69.6 | 61.1 | – | – | – | – | – | – |
| | Qwen2.5-VL (A) | 59.3 | 88.6 | 71.1 | 17.2 | 71.4 | 27.8 | 86.9 | 73.8 | 79.8 |
| | Qwen2.5-VL (S) | 58.5 | 87.3 | 70.0 | 15.4 | 64.3 | 24.8 | 83.3 | 59.5 | 69.4 |
| WebArena | AER-C | 68.8 | 83.2 | 75.3 | – | – | – | – | – | – |
| | AER-V | 67.6 | 84.0 | 74.9 | – | – | – | – | – | – |
| | Claude 3.7 S. (A) | 69.3 | 89.1 | 77.9 | 11.8 | 28.6 | 16.7 | 84.9 | 88.6 | 86.7 |
| | Claude 3.7 S. (S) | 69.3 | 89.1 | 77.9 | 14.8 | 38.1 | 21.3 | 80.8 | 85.1 | 82.9 |
| | Functional | 79.0 | 53.8 | 64.0 | – | – | – | – | – | – |
| | GPT-4o (A) | 70.2 | 89.1 | 78.5 | 9.6 | 90.5 | 17.4 | 82.9 | 93.9 | 88.1 |
| | GPT-4o (S) | 69.9 | 89.9 | 78.7 | 7.9 | 76.2 | 14.3 | 82.7 | 92.1 | 87.1 |
| | GPT-4o Mini (A) | 63.5 | 90.8 | 74.7 | 10.1 | 71.4 | 17.7 | 70.3 | 39.5 | 50.6 |
| | GPT-4o Mini (S) | 66.9 | 86.6 | 75.5 | 7.5 | 28.6 | 11.9 | 86.7 | 11.4 | 20.1 |
| | Llama 3.3 (A) | 68.2 | 86.6 | 76.3 | 8.6 | 76.2 | 15.4 | 79.7 | 86.0 | 82.7 |
| | NNetNav | 54.3 | 90.8 | 67.9 | – | – | – | – | – | – |
| | Qwen2.5-VL (A) | 63.6 | 94.1 | 75.9 | 6.7 | 28.6 | 10.9 | 84.7 | 63.2 | 72.4 |
| | Qwen2.5-VL (S) | 62.9 | 92.4 | 74.8 | 9.4 | 38.1 | 15.1 | 90.5 | 50.0 | 64.4 |
| WorkArena | AER-C | 100.0 | 81.1 | 89.6 | – | – | – | – | – | – |
| | AER-V | 96.4 | 73.0 | 83.1 | – | – | – | – | – | – |
| | Claude 3.7 S. (A) | 85.0 | 91.9 | 88.3 | 8.3 | 50.0 | 14.3 | 76.0 | 86.4 | 80.8 |
| | Claude 3.7 S. (S) | 85.3 | 78.4 | 81.7 | 0.0 | 0.0 | 0.0 | 77.3 | 77.3 | 77.3 |
| | Functional | 100.0 | 91.9 | 95.8 | – | – | – | – | – | – |
| | GPT-4o (A) | 94.6 | 94.6 | 94.6 | 2.6 | 50.0 | 5.0 | 70.4 | 86.4 | 77.5 |
| | GPT-4o (S) | 93.8 | 81.1 | 87.0 | 5.3 | 100.0 | 10.0 | 72.0 | 81.8 | 76.6 |
| | GPT-4o Mini (A) | 84.2 | 86.5 | 85.3 | 3.3 | 50.0 | 6.2 | 66.7 | 36.4 | 47.1 |
| | GPT-4o Mini (S) | 90.3 | 75.7 | 82.4 | 0.0 | 0.0 | 0.0 | 100.0 | 18.2 | 30.8 |
| | Llama 3.3 (A) | 94.3 | 89.2 | 91.7 | 0.0 | 0.0 | 0.0 | 81.8 | 81.8 | 81.8 |
| | NNetNav | 77.3 | 91.9 | 83.9 | – | – | – | – | – | – |
| | Qwen2.5-VL (A) | 87.2 | 91.9 | 89.5 | 0.0 | 0.0 | 0.0 | 100.0 | 59.1 | 74.3 |
| | Qwen2.5-VL (S) | 93.8 | 81.1 | 87.0 | 0.0 | 0.0 | 0.0 | 86.7 | 59.1 | 70.3 |
| WorkArena++ | AER-C | 66.7 | 42.3 | 51.8 | – | – | – | – | – | – |
| | AER-V | 59.3 | 30.8 | 40.5 | – | – | – | – | – | – |
| | Claude 3.7 S. (A) | 66.7 | 62.7 | 64.7 | 17.1 | 38.9 | 23.7 | 87.5 | 97.4 | 92.2 |
| | Claude 3.7 S. (S) | 66.7 | 50.0 | 57.1 | 13.6 | 61.1 | 22.2 | 87.3 | 98.9 | 92.8 |
| | Functional | 83.3 | 38.5 | 52.6 | – | – | – | – | – | – |
| | GPT-4o (A) | 63.0 | 55.8 | 59.2 | 5.5 | 100.0 | 10.3 | 85.6 | 98.5 | 91.6 |
| | GPT-4o (S) | 59.6 | 53.8 | 56.6 | 5.5 | 100.0 | 10.4 | 84.5 | 98.2 | 90.8 |
| | GPT-4o Mini (A) | 49.4 | 76.9 | 60.1 | 5.2 | 83.3 | 9.7 | 87.8 | 52.9 | 66.1 |
| | GPT-4o Mini (S) | 54.8 | 65.4 | 59.6 | 4.6 | 33.3 | 8.1 | 96.5 | 20.2 | 33.4 |
| | Llama 3.3 (A) | 62.7 | 61.5 | 62.1 | 4.7 | 83.3 | 8.9 | 86.7 | 93.8 | 90.1 |
| | NNetNav | 43.2 | 78.8 | 55.8 | – | – | – | – | – | – |
| | Qwen2.5-VL (A) | 60.3 | 78.8 | 68.3 | 7.5 | 77.8 | 13.7 | 91.9 | 79.0 | 85.0 |
| | Qwen2.5-VL (S) | 64.4 | 73.1 | 68.5 | 6.4 | 77.8 | 11.8 | 93.2 | 75.7 | 83.6 |

Table 11: Finegrained results by benchmark and judge for Claude 3.7 Sonnet agent. We report the precision (P) as the primary metric, and F1 and recall (R) as the auxiliary metrics.

| Benchmark | Judge | Success | | | Side Effect | | | Repetition | | |
|---|---|---|---|---|---|---|---|---|---|---|
| | | P | R | F1 | P | R | F1 | P | R | F1 |
| AssistantBench | AER-C | 83.3 | 62.5 | 71.4 | – | – | – | – | – | – |
| | AER-V | 83.3 | 62.5 | 71.4 | – | – | – | – | – | – |
| | Claude 3.7 S. (A) | 87.5 | 87.5 | 87.5 | 7.1 | 33.3 | 11.8 | 70.1 | 94.0 | 80.3 |
| | Claude 3.7 S. (S) | 71.4 | 62.5 | 66.7 | 7.7 | 33.3 | 12.5 | 70.8 | 92.0 | 80.0 |
| | Functional | 25.0 | 12.5 | 16.7 | – | – | – | – | – | – |
| | GPT-4o (A) | 77.8 | 87.5 | 82.4 | 3.2 | 100.0 | 6.3 | 68.6 | 96.0 | 80.0 |
| | GPT-4o (S) | 77.8 | 87.5 | 82.4 | 3.2 | 100.0 | 6.3 | 67.1 | 94.0 | 78.3 |
| | GPT-4o Mini (A) | 80.0 | 100.0 | 88.9 | 2.7 | 66.7 | 5.2 | 59.3 | 32.0 | 41.6 |
| | GPT-4o Mini (S) | 80.0 | 100.0 | 88.9 | 3.4 | 33.3 | 6.3 | 90.0 | 18.0 | 30.0 |
| | Llama 3.3 (A) | 75.0 | 75.0 | 75.0 | 3.4 | 100.0 | 6.6 | 65.8 | 96.0 | 78.0 |
| | NNetNav | 20.8 | 62.5 | 31.2 | – | – | – | – | – | – |
| | Qwen2.5-VL (A) | 72.7 | 100.0 | 84.2 | 0.0 | 0.0 | 0.0 | 73.8 | 62.0 | 67.4 |
| | Qwen2.5-VL (S) | 70.0 | 87.5 | 77.8 | 5.7 | 66.7 | 10.5 | 72.2 | 52.0 | 60.5 |
| VisualWebArena | AER-C | 56.0 | 70.9 | 62.6 | – | – | – | – | – | – |
| | AER-V | 61.2 | 79.7 | 69.2 | – | – | – | – | – | – |
| | Claude 3.7 S. (A) | 61.0 | 77.2 | 68.2 | 16.7 | 35.7 | 22.7 | 78.7 | 96.8 | 86.8 |
| | Claude 3.7 S. (S) | 64.8 | 74.7 | 69.4 | 17.1 | 42.9 | 24.5 | 77.8 | 97.6 | 86.6 |
| | Functional | 85.2 | 58.2 | 69.2 | – | – | – | – | – | – |
| | GPT-4o (A) | 63.0 | 86.1 | 72.7 | 12.4 | 89.3 | 21.8 | 75.2 | 98.4 | 85.2 |
| | GPT-4o (S) | 60.7 | 82.3 | 69.9 | 12.4 | 92.9 | 21.9 | 72.7 | 99.2 | 83.9 |
| | GPT-4o Mini (A) | 57.9 | 83.5 | 68.4 | 10.7 | 64.3 | 18.4 | 74.4 | 46.0 | 56.9 |
| | GPT-4o Mini (S) | 57.4 | 73.4 | 64.4 | 11.2 | 35.7 | 17.1 | 87.1 | 21.4 | 34.4 |
| | Llama 3.3 (A) | 59.6 | 74.7 | 66.3 | 12.0 | 82.1 | 21.0 | 74.4 | 92.1 | 82.3 |
| | NNetNav | 54.5 | 69.6 | 61.1 | – | – | – | – | – | – |
| | Qwen2.5-VL (A) | 59.3 | 88.6 | 71.1 | 17.2 | 71.4 | 27.8 | 86.9 | 73.8 | 79.8 |
| | Qwen2.5-VL (S) | 58.5 | 87.3 | 70.0 | 15.4 | 64.3 | 24.8 | 83.3 | 59.5 | 69.4 |
| WebArena | AER-C | 68.8 | 83.2 | 75.3 | – | – | – | – | – | – |
| | AER-V | 67.6 | 84.0 | 74.9 | – | – | – | – | – | – |
| | Claude 3.7 S. (A) | 69.3 | 89.1 | 77.9 | 11.8 | 28.6 | 16.7 | 84.9 | 88.6 | 86.7 |
| | Claude 3.7 S. (S) | 69.3 | 89.1 | 77.9 | 14.8 | 38.1 | 21.3 | 80.8 | 85.1 | 82.9 |
| | Functional | 79.0 | 53.8 | 64.0 | – | – | – | – | – | – |
| | GPT-4o (A) | 70.2 | 89.1 | 78.5 | 9.6 | 90.5 | 17.4 | 82.9 | 93.9 | 88.1 |
| | GPT-4o (S) | 69.9 | 89.9 | 78.7 | 7.9 | 76.2 | 14.3 | 82.7 | 92.1 | 87.1 |
| | GPT-4o Mini (A) | 63.5 | 90.8 | 74.7 | 10.1 | 71.4 | 17.7 | 70.3 | 39.5 | 50.6 |
| | GPT-4o Mini (S) | 66.9 | 86.6 | 75.5 | 7.5 | 28.6 | 11.9 | 86.7 | 11.4 | 20.1 |
| | Llama 3.3 (A) | 68.2 | 86.6 | 76.3 | 8.6 | 76.2 | 15.4 | 79.7 | 86.0 | 82.7 |
| | NNetNav | 54.3 | 90.8 | 67.9 | – | – | – | – | – | – |
| | Qwen2.5-VL (A) | 63.6 | 94.1 | 75.9 | 6.7 | 28.6 | 10.9 | 84.7 | 63.2 | 72.4 |
| | Qwen2.5-VL (S) | 62.9 | 92.4 | 74.8 | 9.4 | 38.1 | 15.1 | 90.5 | 50.0 | 64.4 |
| WorkArena | AER-C | 100.0 | 81.1 | 89.6 | – | – | – | – | – | – |
| | AER-V | 96.4 | 73.0 | 83.1 | – | – | – | – | – | – |
| | Claude 3.7 S. (A) | 85.0 | 91.9 | 88.3 | 8.3 | 50.0 | 14.3 | 76.0 | 86.4 | 80.8 |
| | Claude 3.7 S. (S) | 85.3 | 78.4 | 81.7 | 0.0 | 0.0 | 0.0 | 77.3 | 77.3 | 77.3 |
| | Functional | 100.0 | 91.9 | 95.8 | – | – | – | – | – | – |
| | GPT-4o (A) | 94.6 | 94.6 | 94.6 | 2.6 | 50.0 | 5.0 | 70.4 | 86.4 | 77.5 |
| | GPT-4o (S) | 93.8 | 81.1 | 87.0 | 5.3 | 100.0 | 10.0 | 72.0 | 81.8 | 76.6 |
| | GPT-4o Mini (A) | 84.2 | 86.5 | 85.3 | 3.3 | 50.0 | 6.2 | 66.7 | 36.4 | 47.1 |
| | GPT-4o Mini (S) | 90.3 | 75.7 | 82.4 | 0.0 | 0.0 | 0.0 | 100.0 | 18.2 | 30.8 |
| | Llama 3.3 (A) | 94.3 | 89.2 | 91.7 | 0.0 | 0.0 | 0.0 | 81.8 | 81.8 | 81.8 |
| | NNetNav | 77.3 | 91.9 | 83.9 | – | – | – | – | – | – |
| | Qwen2.5-VL (A) | 87.2 | 91.9 | 89.5 | 0.0 | 0.0 | 0.0 | 100.0 | 59.1 | 74.3 |
| | Qwen2.5-VL (S) | 93.8 | 81.1 | 87.0 | 0.0 | 0.0 | 0.0 | 86.7 | 59.1 | 70.3 |
| WorkArena++ | AER-C | 66.7 | 42.3 | 51.8 | – | – | – | – | – | – |
| | AER-V | 59.3 | 30.8 | 40.5 | – | – | – | – | – | – |
| | Claude 3.7 S. (A) | 66.7 | 62.7 | 64.7 | 17.1 | 38.9 | 23.7 | 87.5 | 97.4 | 92.2 |
| | Claude 3.7 S. (S) | 66.7 | 50.0 | 57.1 | 13.6 | 61.1 | 22.2 | 87.3 | 98.9 | 92.8 |
| | Functional | 83.3 | 38.5 | 52.6 | – | – | – | – | – | – |
| | GPT-4o (A) | 63.0 | 55.8 | 59.2 | 5.5 | 100.0 | 10.3 | 85.6 | 98.5 | 91.6 |
| | GPT-4o (S) | 59.6 | 53.8 | 56.6 | 5.5 | 100.0 | 10.4 | 84.5 | 98.2 | 90.8 |
| | GPT-4o Mini (A) | 49.4 | 76.9 | 60.1 | 5.2 | 83.3 | 9.7 | 87.8 | 52.9 | 66.1 |
| | GPT-4o Mini (S) | 54.8 | 65.4 | 59.6 | 4.6 | 33.3 | 8.1 | 96.5 | 20.2 | 33.4 |
| | Llama 3.3 (A) | 62.7 | 61.5 | 62.1 | 4.7 | 83.3 | 8.9 | 86.7 | 93.8 | 90.1 |
| | NNetNav | 43.2 | 78.8 | 55.8 | – | – | – | – | – | – |
| | Qwen2.5-VL (A) | 60.3 | 78.8 | 68.3 | 7.5 | 77.8 | 13.7 | 91.9 | 79.0 | 85.0 |
| | Qwen2.5-VL (S) | 64.4 | 73.1 | 68.5 | 6.4 | 77.8 | 11.8 | 93.2 | 75.7 | 83.6 |

Table 12: Finegrained results by benchmark and judge for GPT-4o agent. We report the precision (P) as the primary metric, and F1 and recall (R) as the auxiliary metrics.

```
System Prompt Template

You are a Language Model specialized in judging the performance of web agents in web
navigation tasks. For a certain website, you are given the goal of a navigation task,
the current URL of the webpage, the actions taken by the agent, and the thought
process of the agent. Additionally, you will have access to the first and last
screenshots, as well as the last accessibility tree. Your task is to answer several
questions about the agent's performance in the task.

Question 1: Was the sequence of actions below successful in achieving the goal?
Choices: <success>Successful</success>, <success>Unsuccessful</success>

Question 2: Did the agent perform unnecessary actions that could lead to unintended
side effects?
Choices: <side>Yes</side>, <side>No</side>

Question 3: Did the agent perform the task optimally, by only performing necessary
actions and avoiding unnecessary ones?

Choices: <optimal>1. Complete Failure</optimal>, <optimal>2. Suboptimal</optimal>,
<optimal>3. Somewhat Optimal</optimal>, <optimal>4. Completely Optimal</optimal>

Question 4: Did the agent loop through a sequence of actions that did not make
progress towards the goal?
Choices: <loop>Yes</loop>, <loop>No</loop>

You should provide your reasoning process for each question, and you can refer to
the screenshots and the accessibility tree to support your answers. Your answer
must follow the following format:

<reasoning>your reasoning here</reasoning>
<success>answer</success>
<side>answer</side>
<optimal>answer</optimal>
<loop>answer</loop>
```

Figure 5: System Prompt Template used for the simplified judge

```
User Prompt Template

The user goal is: {goal}
The agent performed the following actions:
-----
Step: {step_number}
URL: {url}
Action: {action}
Reasoning: {reasoning}
-----
...
-----
The last accessibility tree is:
{axtree}

Here is the screenshot of the last step.
{screenshot}

Provide your reasoning and answer the four questions from the system prompt, using
the specified format.
```

Figure 6: User Prompt Template used for the simplified judge

