# OpenReview forum: "AgentRewardBench: Evaluating Automatic Evaluations of Web Agent Trajectories"
_colmweb.org/COLM/2025/Conference — COLM 2025_

### Official Review · Reviewer_2xan · 2025-05-11

**Rating:** 7
**Confidence:** 2
**Ethics Flag:** 1

**Summary:**

The paper propose the AgentRewardBench a benchmark to assess the effectiveness of LLM judges for evaluating web agents. The benchmark has1300 trajectories collected based on popular LLMs across 5 diverse web environments. The benchmark uses a simplified judge design that can receive either the accessibility tree or a screenshot as the representation of the final state. The author also shared several important findings from the evaluation.

## Quality
The paper constructs a benchmark dataset comprising 1300 annotated trajectories from multiple web agents across diverse benchmarks. It also leverages human annotations to provide high-quality ground-truth labels. The paper also evaluates several large language models as judges, comparing them with both expert annotations and conventional rule-based evaluations. In the error analysis, it also provides comprehensive analysis and insights.

However, it does not cover the information related to the computational overhead or scalability.

## Clarity
The paper is well written with clear structure and motivation.

## Originality
The introduces the first benchmark about evaluating the reliability and accuracy of LLM judges for web agent trajectories. It provides the limitations of rule-based evaluation methods compared to LLM-based evaluation, contributing original empirical findings.

However, this paper only has incremental innovation.  Even the benchmark itself is novel, the underlying concept is not new.

**Reasons To Accept:**

## High quality
The paper constructs a benchmark dataset comprising 1300 annotated trajectories from multiple web agents across diverse benchmarks. It also leverages human annotations to provide high-quality ground-truth labels. The paper also evaluates several large language models as judges, comparing them with both expert annotations and conventional rule-based evaluations. In the error analysis, it also provides comprehensive analysis and insights.


## Clear presentation
The paper is well written with clear structure and motivation.

**Reasons To Reject:**

## Limited innovation
This paper has incremental innovation. The benchmark itself is novel, however the underlying concept is an incremental advancement.

---

> ### Author Response · Authors · 2025-06-02
>
> Dear reviewer,
>
> We thank you for their positive feedback on AgentRewardBench, particularly recognizing its high-quality dataset, rigorous human annotations, comprehensive LLM judge evaluation, and clear presentation.
>
> We would like to address your concern:
>
> > The benchmark itself is novel, however the underlying concept is an incremental advancement.
>
> We would like to point out that our work is far from incremental. AgentRewardBench is the first dedicated benchmark to rigorously assess LLM judges specifically for web agent trajectories. We made several key contributions. (1) Novelty of the Benchmark: Creating a dataset of 1300 annotated trajectories across 5 diverse web environments with high-quality expert annotations is a substantial undertaking. This enables standardized, reproducible research into automatic evaluation for web agents. (2) Addressing Existing Limitations: We empirically show the shortcomings of traditional rule-based evaluations, highlighting the need for nuanced automatic evaluations. AgentRewardBench provides the means to systematically validate this approach. (3) Specific Focus on Web Agents: Evaluating web agents is uniquely challenging. Our benchmark is tailored to these challenges, considering representations like accessibility trees and screenshots, distinguishing it from general LLM evaluation studies.
>
> Therefore, while the LLM-as-a-judge concept may extend existing ideas, creating AgentRewardBench as the first benchmark for this complex domain is a crucial advancement. It provides a foundation for future research to evaluate and improve judges for web agents.

---

### Official Review · Reviewer_9MNf · 2025-05-19

**Rating:** 7
**Confidence:** 3
**Ethics Flag:** 1

**Summary:**

This paper presents AgentRewardBench, a new benchmark for evaluating reward models for web agent trajectories. This benchmark consists of over 1300 trajectories generated by multiple agents across five web agent benchmarks, with each trajectory manually labeled by human experts. This work systematically evaluates 12 LLM-based judges and existing rule-based systems, providing detailed comparative analysis, ablation studies, and error cases. This work is timely for the community to understand the current status of reward model for web agents and further develop and evaluate these reward models. The paper is well-written.

**Questions To Authors:**

* Training models towards a reward model is an exciting application of the annotated reward labels provided in this work. Have the authors explored training a model using the collected dataset? If so, how does this influence model or agent performance? Any results or insights from such experiments would be valuable to share.

**Reasons To Accept:**

* This paper studies a very important problem for the agent community: building and evaluating reward models for agent trajectories, which can potentially benefit a lot of relevant research, like reinforcement learning, inference time search, etc.

* This work presents a thorough and systematic study regarding the performance of different judges across five web agent benchmarks under a unified setting, along with a details analysis and insights.

* The annotation protocols and the released dataset can provide valuable resources for the community to enable further research, especially on scaling up the reward label scale and further evaluating the reward model with training.

**Reasons To Reject:**

* The evaluation of judgment methods appears to be missing an important baseline of the Agent-as-Judge approach. This method is recognized as a powerful method for the automatic evaluation of agent trajectories. Including results for this baseline would strengthen this study and provide a more comprehensive comparison.

---

> ### Author Response · Authors · 2025-06-02
>
> Dear reviewer,
>
> Thank you for your thorough review of our work. We would like to address your concerns and questions below.
>
> ---
>
> ## Including “Agent-as-Judge” as a baseline
>
> > The evaluation of judgment methods appears to be missing an important baseline of the Agent-as-Judge approach. This method is recognized as a powerful method for the automatic evaluation of agent trajectories. Including results for this baseline would strengthen this study and provide a more comprehensive comparison.
>
> Thanks for pointing us to this work, which we will include in the related works section for camera ready.
>
> We agree that the Agent-as-Judge (AaJ) approach is a promising method for evaluating agent trajectories. However, AaJ assumes the judge can interact with the environment after the agent finishes executing actions, but this isn’t feasible when the environment state cannot be preserved or shareable across agents (including AssistantBench, WebArena, WorkArena, etc.). However, we agree that Agent-as-Judge could be useful for web environments where the state can be preserved and made available to the judge easily.
>
> After the anonymity period, we will reach out to the AaJ authors to explore ways to adapt their framework for cases like ours.
>
> ---
>
> ## Questions
>
> > Training models towards a reward model is an exciting application of the annotated reward labels provided in this work. Have the authors explored training a model using the collected dataset? If so, how does this influence model or agent performance? Any results or insights from such experiments would be valuable to share.
>
> We used the entirety of the collected trajectories as part of the benchmark. However, we believe that collecting additional trajectories for training reward models is an exciting direction for future research. Our benchmark will be the perfect fit for evaluating the performance of such finetuned reward models.

---

> > ### Comment · Reviewer_9MNf · 2025-06-10
> > **Thanks for your response**
> >
> > Dear Authors,
> >
> > I understand the limitations regarding the Agent-as-Judge baseline and appreciate your plan to explore its applicability in the future. I also agree that your dataset provides a strong foundation for training and evaluating reward models going forward. I will maintain my scores.

---

> ### Comment · Area_Chair_zdHF · 2025-06-08
>
> Hi reviewer! A quick reminder that the discussion period ends on 6/10. Please make sure to read and respond to the authors' rebuttals -- It'd mean a lot to the authors who have put in a lot of efforts into their work!

---

### Official Review · Reviewer_NwGC · 2025-05-26

**Rating:** 7
**Confidence:** 4
**Ethics Flag:** 1

**Summary:**

The authors introduce AgentRewardBench, which is a new benchmark for evaluating the effectiveness of LLM judges that assess the success of web agents. While there are existing benchmark datasets that evaluate LLM-based judges for other tasks, this seems to be the first one that evaluates LLM judges for web agents. The dataset contains 1302 trajectories generated by 4 different LLMs across 5 popular web agent benchmarks and annotated by human experts on three dimensions: success/failure, side effects and repetitiveness.

These are some of their findings:
- The official rule-based evaluations of these 5 benchmarks can be inflexible and tend to under report the success rate
- None of the existing judges they evaluated exceeded 70% precision compared to expert annotations
- Existing LLM judges fail due to some common error categories such as misleading web-agent reasoning, failing to report missed instruction details, grounding mismatch (judge does not have access to screenshots, only the web agent's thought process) etc.

The paper is well-written and structured. The authors clearly describe the problem statement, proposed benchmark, experimental details, results etc. along with informative figures and tables. They also include an in-depth evaluation of the results and qualitative error analysis. The impact of this paper is potentially high, given that it's based on some existing popular web agent benchmarks. I believe the paper should be accepted.

**Questions To Authors:**

- Nitpick: In the Appendix, Line 512 mentions AgentLab, before you actually describe what it is in the next paragraph (Line 519-522).
- Was the code/dataset not included as supplementary data?
- Did you consider annotating for more granular success/failure signals, such as 'partially successful,' or did you find it unnecessary?

**Reasons To Accept:**

- The paper is well-written and structured
- Useful first-of-its-kind benchmark for evaluating LLM judges focused on web agent evaluation. Tackles the under-explored area of evaluating the evaluators themselves.
- The tasks and trajectories are based on five popular web agent benchmarks, so will prove valuable to many users.
- Useful analysis of the current limitations of existing LLM judges as well as official rule-based evaluations.
- Qualitative error analysis highlighting common error categories where existing LLM judges fail.
- The authors provide substantial details regarding their methodology, including agent platform, annotation processes, example prompts etc. which helps with reproducibility and verifiability of their findings.
- This benchmark will help improve automatic evaluation methods, which in turn will be useful for synthesizing trajectories for fine-tuning, reward models for reinforcement learning etc.

**Reasons To Reject:**

Some minor points:

Related works should have focused more on LLM judges and existing benchmarks that evaluate them rather than web agents / environments / trajectory synthesis.

The authors mention that "The team of annotators consisted of 6 experts who work on web agents research" ... which suggests that the annotators were probably the authors themselves or close colleagues from the same research group. This does have a few disadvantages:
 --(1) these researchers might interpret success or failure in a way that's somewhat different from the more typical user
 --(2) A small group of closely working experts might develop a very specific, shared understanding of the annotation criteria that isn't explicitly documented and might not be easily replicable by an external group. The authors state that annotators resolved disagreements/ambiguous situations through discussion. The resulting consensus should be compiled and included in the appendix as annotator guidelines.
  On the other hand, having experts with deep domain knowledge as annotators is also beneficial (focusing on things that matter rather than quirks of web agent trajectories), since questions related to trajectory optimality and side effects of unnecessary actions might be harder for the average user to answer.

Overall, the strengths of this paper outweigh these minor points.

---

> ### Author Response · Authors · 2025-06-02
>
> Dear reviewer,
>
> Thank you for your insightful review, and for highlighting that it is a first-of-its-kind benchmark addressing an under-explored area of web agent evaluation. We agree that it will help improve automatic evaluation methods, and we are glad you appreciated the substantial details from our methodology and annotation processes.
>
> We would like to now address your suggestions on how to improve the paper.
>
> ---
>
> ## Suggestions
>
> > Related works should have focused more on LLM judges and existing benchmarks that evaluate them rather than web agents / environments / trajectory synthesis.
>
> Thank you for this suggestion. We have already included the following papers about LLM judges and benchmarks: LLM-as-a-judge (Zheng et al., 2023), AER (Pan et al., 2024), and RewardBench (Lambert et al., 2024).
> We will add more past and concurrent works on the topic of LLM judges and evaluation, including Agent-as-a-Judge (Zhuge et al.). We are happy to incorporate your suggestions as well.
>
> > (1) these researchers might interpret success or failure in a way that's somewhat different from the more typical user
> > (2) A small group of closely working experts might develop a very specific, shared understanding of the annotation criteria that isn't explicitly documented and might not be easily replicable by an external group.
> >
> > …
> >
> > The resulting consensus should be compiled and included in the appendix as annotator guidelines.
>
> Thank you for this insightful suggestion. In addition to including the UI and annotation instructions (appendix A.2 and Figure 4), we recognize the value of discussing the resolution process and potential shared understanding that is not already present in the instructions. A salient reason for mismatch is ambiguity of the instructions (the instruction might mention “buy an black t-shirt”, but does not specify if it is fully black or can have other graphics). In such cases, annotators are advised to go for the most lenient option. We will create a new section, A.4, that will discuss this.
>
> > having experts with deep domain knowledge as annotators is also beneficial (focusing on things that matter rather than quirks of web agent trajectories), since questions related to trajectory optimality and side effects of unnecessary actions might be harder for the average user to answer.
> >
> > Overall, the strengths of this paper outweigh these minor points.
>
> We agree that it is hard for the average user to analyze web trajectories; this is the primary reason why we decided to leverage expert annotations.
>
> ---
>
> ## Questions
>
> > Q1: Nitpick: In the Appendix, Line 512 mentions AgentLab…
> Thanks for catching this, we will fix this in the camera-ready.
>
> > Q2: Was the code/dataset not included as supplementary data?
>
> Our code and dataset are available and we will share the link after the anonymity period.
>
> > Q3: Did you consider annotating for more granular success/failure signals?
>
> We initially considered step-level reward annotations, but found them costly to obtain and not especially helpful: current judges already struggle with trajectory-level evaluation. Instead, we focused on annotating specific failure modes like repetitions and side effects, which provided more actionable insights. Step-level signals remain valuable, and we view them as an important direction for future work given better judge capabilities and more resources.

---

> > ### Comment · Reviewer_NwGC · 2025-06-04
> >
> > Thank you for your response. I appreciate the additions you plan to make to the paper including more related works on the topic of LLM judges and the new appendix A.4 containing additional annotator instructions/guidelines. Overall I believe this is a good paper and I stand by my positive review.

---

### Official Review · Reviewer_NzzY · 2025-05-27

**Rating:** 5
**Confidence:** 4
**Ethics Flag:** 1

**Summary:**

This paper introduces a benchmark, AGENTREWARDBENCH, to assess the effectiveness of LLM judges for evaluating web agents. LLM judges evaluate web agent in terms of success, side effects, and repetitiveness of the agent. In this paper, 12 LLM judges are evaluated on this benchmark. It is found that no single LLM excels across all tasks and rule-based evaluation tend to underestimate the capability of web agents.

**Reasons To Accept:**

1. The dataset introduced by this paper can be of interest to the research community, e.g. in terms of its scale and diversity: it consists of 1300 trajectories produced by 4 popular LLM agents on 5 diverse web environments, with expert annotations of the ground truth judgement.

2. Multiple aspects are used to evaluate LLM agents comprehensively, including success, side effects and repetition cycle. Not only is the execution success, but also the effectiveness of LLM agents, is taken into consideration.

3. The paper is clearly structured and easy to read. The multiple diagrams or illustrations help reads understand the paper better. The mathematical description of the assessment framework (including trajectory definition, annotation design, judge model, etc.) is rigorous.

**Reasons To Reject:**

1. In result reporting, the paper uses precision as the primary metrics, and recall and F1 as the secondary metrics. However, the recall can be as important as precision in many RL applications. For example, if an LLM judge is employed as a reward model, it is important that incorrect trajectories are not noted as correct to give LLM agents wrong signals.

2. The "error analysis" section is not very complete: it only includes only failure cases on the "success / completion" dimension of LLM agents, missing the other two dimensions (side effect, repetition) included in the benchmark.

---

> ### Author Response · Authors · 2025-06-02
>
> Dear reviewer,
>
> We thank you for highlighting the scale and diversity of our benchmark, as well as the inclusion of expert annotations, which we agree will be of interest to the research community. We are grateful that you recognize the comprehensiveness and rigour of our evaluation framework, as well as the clarity of our writing.
>
> We would like to address your concerns below.
>
> ---
>
> ## Importance of recall compared to precision
>
> > the recall can be as important as precision in many RL applications. For example, if an LLM judge is employed as a reward model, it is important that incorrect trajectories are not noted as correct to give LLM agents wrong signals.
>
> We agree that “it is important that incorrect trajectories are not noted as correct”. This was indeed our primary motivation for highlighting precision, as the number of incorrect trajectories being judged as correct (false positives) does not influence recall. On lines 240-244 of our submission, we explain why we use the precision score:
>
> > … we are interested in increasing the number of true positives (actual successful trajectories) while reducing the number of false positives (failed trajectories added to the dataset due to poor LLM judgments).  For reward modeling, we also want to prioritize true positives since they are the primary signals for many RL algorithms, while false positives should be minimized to avoid introducing noise to the loss function.
>
> As we can see from the precision score, defined as tp / (tp + fp), where tp is the true positive and fp is the false positive, maximizing this value will result in fewer incorrect trajectories being labeled as correct. **Thus, precision precisely addresses your concern.**
>
> That said, we recognize the value of reporting recall and F1 for completeness. **That is why we have already reported both metrics alongside precision.** For instance, we report in table 1 that screen-based Claude 3.7 Sonnet achieves a precision of 69.4, but also an F1 score of 72.7 and recall of 76.3; the finegrained recall and F1 scores are included in the Appendix tables 7-8.
>
> It would not require additional effort to move the results from table 8 to table 1, if you believe this would be a valuable addition. We are also open to giving more emphasis to the F1 scores we reported, if you believe there are strong reasons to highlight other properties of the recall component.
>
> ---
>
> ## Inclusion of side effect and repetition in error analysis
>
> > The "error analysis" section is not very complete: it only includes only failure cases on the "success / completion" dimension of LLM agents, missing the other two dimensions (side effect, repetition) included in the benchmark.
>
> We emphasize the completion dimension in the error analysis in order to remain focused on the discussions from sections 4.2 to 5. However, we are happy to add supplementary error analyses that would cover the side effect and repetition dimension, which will be a straightforward addition to the paper.
>
> ---
>
> We hope this clarification addresses your concerns and encourages a fresh evaluation of our work. Given that we clarified the nature of the precision score, and that further error analyses will be added to the camera ready version, we would like you to reconsider your score.
>
> We are happy to discuss further.

---

### Decision · Program_Chairs · 2025-07-08

**Decision:**

Accept

**Comment:**

AgentRewardBench introduces a 1.3 k-trajectory benchmark to see how well LLM “judges” agree with humans on web-agent outcomes. Reviewers largely align on the same strengths (useful dataset + clear study) and note similar weaknesses (metric emphasis, missing baselines/docs):

Strengths
1. The dataset is large and diverse (NzzY, NwGC, 9MNf, 2xan)
2. The paper is the first to target LLM judges for web agents (NwGC, 9MNf)
3. Evaluation is thorough: 12 judges vs rule-based baselines + error analysis (NwGC, 9MNf, 2xan)
4. The paper provided enough detailed methodology & prompts that supports reproducibility (NwGC)

Weaknesses
1. The evaluation is a bit biased towards certain metrics and cases (NzzY), e.g., metrics favors precision over recall, error analysis focuses on success demisions. Authors explained the reason and expand appendices with more analysis.
2. The paper missed some related work on evlauation of judges (NwGC, 9MNf). Authors mentioned they would add papers Agent-as-Judge, RewardBench, etc. & also mentioned AsJ is hard to be adapted to this setting.
3. Data distribution warrants more discussion -- e.g. Annotator pool seems small and homogeneous (NwGC). Authors mentioned they would create new appendices for this.
4. There were also some concerns on the conceptual novelty (2xan), though the reviewer also agreed that the dataset itself is a novel contribution.

Overall most reviewers find this to be a valuable paper and a useful benchmark for the community. We hope that author could make the promised changes to the revised paper and release the code and the data!